# Eyes Wide Open:
# Ego Proactive Video-LLM for Streaming Video

**Yulin Zhang**[1]    **Cheng Shi**[3]    **Yang Wang**[1]    **Sibei Yang**[2†]

[1]ShanghaiTech University    [2]School of Computer Science and Engineering, Sun Yat-sen University
[3]School of Computing and Data Science, The University of Hong Kong
**Project Page:** https://zhangyl4.github.io/publications/eyes-wide-open/

## Abstract

Envision an AI capable of functioning in human-like settings, moving beyond mere observation to actively understand, anticipate, and proactively respond to unfolding events. Towards this vision, we focus on the innovative task where, *given ego-streaming video input, an assistant proactively answers diverse, evolving questions at the opportune moment, while maintaining synchronized perception and reasoning.* This task embodies three key properties: (1) Proactive Coherence, (2) Just-in-Time Responsiveness, and (3) Synchronized Efficiency. To evaluate and address these properties, we first introduce ESTP-Bench (Ego Streaming Proactive Benchmark) alongside the ESTP-F1 metric—a novel framework designed for their rigorous assessment. Secondly, we propose a comprehensive technical pipeline to enable models to tackle this challenging task. This pipeline comprises: (1) a data engine, (2) a multi-stage training strategy, and (3) a proactive dynamic compression technique. Our proposed model effectively addresses these critical properties while outperforming multiple baselines across diverse online and offline benchmarks.

## 1 Introduction

Imagine an AI assistant that follows you through your day—assembling furniture, searching for misplaced keys [3, 15], or preparing a meal [46, 44]—not just watching, but understanding, anticipating [54], and responding proactively when needed as events unfold. To function in such human-like settings, where visual input is egocentric and continuously streaming, and user needs shift from moment to moment, the assistant must go beyond passive observation. It should be able to interpret the present, anticipate what comes next, and respond at exactly the right moment, all in real time.

As a first step toward this vision, we narrow our focus to perception and understanding in egocentric streaming video, with a particular emphasis on the following innovative task: *Given ego-streaming video input, the assistant proactively answers to diverse and evolving questions at the right moment, while seeing and thinking in sync,* as shown in Fig. 1. This task relies on three key properties:

- Proactive Coherence: handling diverse question types, responding even when answers depend on future visual streams (proactivity), and maintaining contextual consistency across related questions. In ego-streaming scenarios, questions often go beyond the current frame, referencing future events or past observations. As shown in Fig. 1, the segment of the conversation highlighted in green is contextually dependent on the content within the segment highlighted in purple. Such queries require temporal integration of past and present information, followed by proactive answering as relevant visual evidence emerges.
- Just-in-Time Responsiveness: determining when to answer based on visual readiness, neither too soon nor too late, and only when necessary. Responding before enough evidence is available can

---

†Corresponding author is Sibei Yang.

39th Conference on Neural Information Processing Systems (NeurIPS 2025).

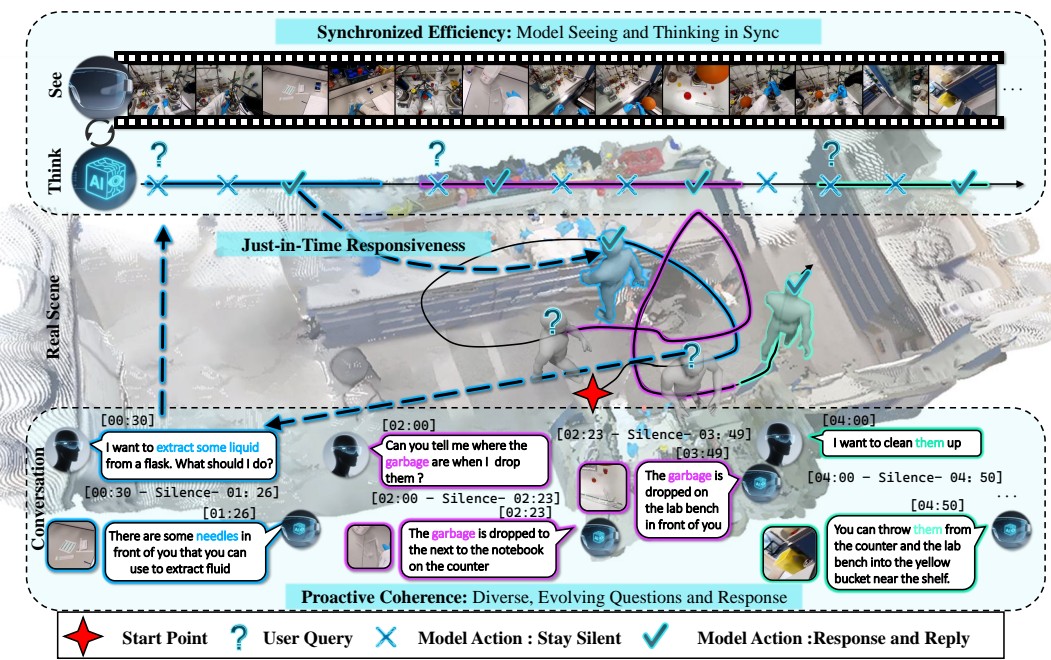

Figure 1: **An illustrative example of the ESTP task.** The figure is structured in three layers: the top layer depicts the model's continuous visual processing and decision-making (*See and Think*), the middle layer shows the real-world egocentric scene with the human's trajectory, and the bottom layer presents the human-model conversation.

lead to mistakes, while answering too late may miss the opportunity to help. Equally important is staying silent when uncertain and avoiding unnecessary repetition. As shown in the blue-highlighted segment of Fig. 1, it is necessary to remain silent until the "face to counter". The assistant must continuously track the evolving visual context and respond at the earliest reliable moment.

- Synchronized Efficiency: ensuring that answering and visual perception proceed in sync without delay. Responses should not come at the cost of missing new visual input; perception and reasoning must remain temporally aligned. Regarding the purple segment depicted in Fig. 1, maintaining synchronization is crucial to prevent missed answers. This requires answering while continuously observing, with zero latency, while also ensuring time and memory efficiency as the number of incoming frames grows over time.

Unfortunately, existing evaluation frameworks [51, 27, 26, 61, 13] and streaming models [5, 52] fall short in supporting or measuring the unified capabilities of proactive, just-in-time, and synchronized reasoning—and often struggle even with some individual aspects. Offline video benchmarks [14, 66, 30, 50, 1] evaluate video LLMs across diverse question types and scenarios, but their offline nature limits the assessment of the three core capabilities essential for online deployment. Recent efforts toward online and streaming benchmarks address this gap by introducing proactive tasks. Nevertheless, as shown in Tab 1, they often offer limited question diversity, lack contextual continuity across queries, and—more importantly—rarely evaluate just-in-time responsiveness or synchronized efficiency. As a result, current online video LLMs remain confined to narrow tasks such as narration or simple question answering, lacking the capacity for continuous, multi-turn understanding. More critically, as illustrated in Fig. 6, these models exhibit poor just-in-time behavior—often generating under-responsive or over-extended answers. Similarly, although recent efforts [51, 33] have begun to address efficiency, they tend to focus solely on accelerating response generation—potentially at the cost of answer accuracy—while overlooking the need to balance perception and answering under synchronized constraints.

As a first step toward addressing these challenges, we introduce *a new Ego STreaming Proactive (ESTP) benchmark and evaluation framework*, specifically designed to capture the demands of the three key properties in streaming video. **For proactive coherence**, all question-answering tasks in the benchmark are proactive in nature: each question can only be answered based on future video streams

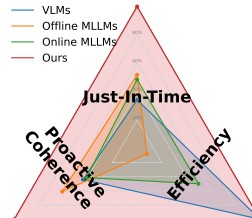

Figure 2: ESTP Triangle of Impossibility shows trade-offs among the three dimensions: Proactive Coherence, Just-in-Time responsiveness, and Efficiency, which are quantified by contextual performance, recall, and FPS.

| Dataset | Ques. Type | Proactive Type | | | JIT Responsiveness Eval. | | | # Ques. |
|---|---|---|---|---|---|---|---|---|
| | | Exp. | Imp. | Cont. | Ans. Turn | Is Prec. | Timeliness | |
| ***Online Benchmark*** | | | | | | | | |
| VStream [61] | OE | ✗ | ✗ | ✗ | S | ✗ | ✗ | 3,500 |
| StreamingBench [27] | MC | ✗ | ✗ | ✗ | S | ✗ | ✗ | 4,500 |
| StreamingBench (PO) [27] | Q-Match | ✔ | ✗ | ✗ | S | ✗ | ✔ | 50 |
| OVO-Bench [26] | MC | ✗ | ✗ | ✗ | S | ✗ | ✗ | 2,814 |
| OVO-Bench (FAR) [26] | C & Q | ✔ | ✗ | ✗ | M | ✔ | ✗ | 1618 |
| MMDuet [51] | OE | ✔ | ✗ | ✗ | M | ✗ | ✔ | 2000 |
| ***Ego Benchmark*** | | | | | | | | |
| EgoPlan [6] | OE | ✗ | ✗ | ✗ | S | ✗ | ✗ | 5,000 |
| EgoPlan2 [35] | OE | ✗ | ✗ | ✗ | S | ✗ | ✗ | 1,300 |
| EgoSDQES [13] | Q-Match | ✔ | ✗ | ✗ | S | ✔ | ✔ | 3,971 |
| ESTP (Ours) | OE | ✔ | ✔ | ✔ | M | ✔ | ✔ | 2264 |

Table 1: Comparison of datasets based on proactive and streaming criteria. This table summarizes datasets by Question Types (Open-Ended (OE); Multiple Choice (MC); Query Matching (Q-Match & Q); and Count (C)), Proactive Types (Explicit (Exp.); Implicit (Imp.); and Contextual (Cont.)), and Just-in-Time (JIT) Responsiveness. Key JIT Responsiveness aspects include Answer Turn (Ans. Turn) (options: Single (S), Multi (M)), Precision (Is Prec.), and Timeliness. The notation '# Ques.' denotes the number of questions.

within one or more specific time intervals. To reflect different levels realistic scenarios, we group them into three types: (1) explicit, grounded in clear visual cues; (2) implicit, requiring reasoning beyond surface observations; and (3) contextual, involving temporally linked questions that demand consistent multi-turn answers. We collect 2,264 questions spanning 14 task types—such as object localization, state change understanding, and intention prediction—across over 100 types of distinct scenarios, including kitchen activities, social interactions, and daily object manipulation. **For just-in-time responsiveness**, we emphasize the importance of response timing: each question are annotated an average of 3.96 valid answer intervals, and a prediction is considered valid only if it falls within the designated window. To assess this, we introduce ESTP-F1, a metric that integrates answer quality, response timing, and temporal precision. Additionally, 46% of questions are contextually linked, requiring coherent responses based on prior questions—highlighting the need to continuously track the evolving stream from past to future and respond at the right moment. **For synchronized efficiency**, we not only evaluate time and memory efficiency and answering accuracy independently, but also assess accuracy under tightly synchronized perception and response—offering a comprehensive perspective on streaming video LLM evaluation.

To address this novel task, *we propose a comprehensive and novel technical pipeline—including a data engine, multi-stage training strategies, and a proactive dynamic compression technique—to enhance the streaming video LLMs. Specifically*,

- **The data engine** automatically generates diverse, multi-turn questions and their corresponding answers to support the demands of continuous and proactive question answering. This involves a three-stage generation pipeline covering (1) one-to-one: using LVLMs to generate captions and extract initial question-answer pairs with a single temporal answer interval; (2) one-to-many: applying RAG to expand each answer into multiple valid intervals; and (3) many-to-many: composing coherent multi-turn questions from related QA pairs.

- **The multi-stage training strategy** is employed to progressively learn: (1) passive interval responsiveness, which provides a basic ability to trigger responses by distinguishing visually similar frames with different response labels, but often results in over-responsiveness even when the correct response interval; (2) Proactive just-in-time responsiveness and accurate answering, which trains the model to actively request high-resolution frames during uncertain timestamps, allowing it to use fine-grained visual details to pinpoint both the correct response moment and the accurate answer; (3) Coherence across multi-turn QA, which enables the model to maintain consistency by reasoning over prior QA history and current context, supporting contextual consistency answering.

- **The proactive dynamic compression** technique fully leverages the streaming nature by applying two levels of token compression based on response likelihood, including: (1) when the model anticipates a potential response, it proactively requests high-resolution inputs to improve the accuracy of perception and answering; (2) Otherwise, it applies a higher compression rate to past content to reduce token usage and improve efficiency; (3) Additionally, once a response is completed, the content preceding its timestamp is further compressed to free up resources without affecting future perception or answering.

In summary, our contributions include: the novel Ego-Streaming Proactive (ESTP) task, distinguished by its three key properties; the ESTP-Bench benchmark and the ESTP-F1 metric for robust evaluation of this task; and a comprehensive and novel technical pipeline, incorporating three key techniques, designed to address the ESTP task. Our results demonstrate that the proposed model effectively overcomes the key challenges posed by this task. Moreover, it demonstrates superior performance by substantially exceeding multiple baselines in diverse online and offline benchmarks.

## 2 Ego Streaming Proactivate Dataset & Benchmark

### 2.1 Data Source and Annotation

**Data Source** is validation set of Ego4D [17, 44] that includes raw annotations such as event narrations and steps for completing consistent goals. Following [28, 5], we filtered out video with missing or uncertain annotations and converted annotations into a natural language format. This process yielded 890 videos, encompassing over 100 distinct scenes and a wide array of human activities, including indoor home environments (e.g., cooking, cleaning), workspaces (e.g., working at desk, labwork, baker), and public areas (e.g., grocery shopping). Furthermore, the videos exhibit rich dynamic diversity, ranging from periods of relative stillness (e.g., observing a static scene) to highly dynamic moments involving rapid manipulation tasks or active locomotion (e.g., cooking, walking).

**Annotation** process follows a two-step procedure. First, initial QA pairs are automatically generated with the assistance of MLLMs [56, 49] and LLMs [10]. Second, these automatically generated questions provided inspiration for annotators, aiding them in identifying valuable instances or formulating question ideas. To ensure diversity of questions, we annotate three proactive types:

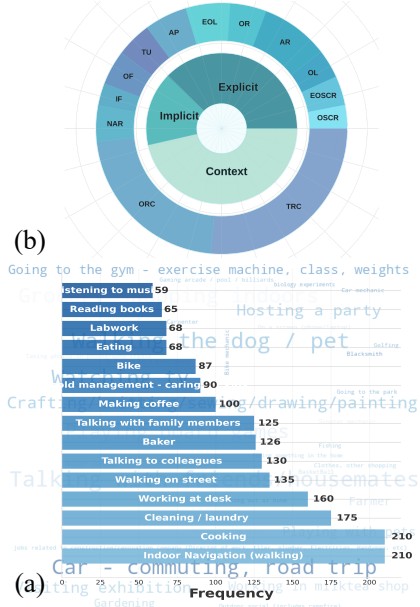

(b)

(a)

Figure 3: (a) Frequency of scenes or activities from which tasks and questions are derived. (b) Proportion of different proactive and question task types.

**(1). *Explicit Proactive Tasks*** are defined as those required to identify and respond to queries by directly leveraging and interpreting visual information present in the input. This category encompasses tasks where the relevant visual cues are explicitly referenced or are central to formulating a correct response. This category is comprised of eight distinct task types: Object Recognition (OR), Attribute Perception (AP), Text-Rich Understanding (TRU), Object Localization (OL), Object State Change (OSC), Ego Object Localization (EOL), Ego Object State Change (EOSC), and Action Recognition (AR). **(2). *Implicit Proactive Tasks*** are defined as those requiring inference and deeper scene understanding that goes beyond immediate, direct observation. This category is comprised of four distinct task types: Object Function Reasoning (OFR), Information Function Reasoning (IFR), Next Action Reasoning (NAR), and Task Understanding (TU). **(3). *Contextual Proactive Tasks*** are defined as those requiring the model to maintain awareness of dialogue history and visual coherence across temporally extended interactions. This category is comprised of two distinct task types: Object Relative Context (ORC) and Task Relative Context (TRC). Fig. 3 illustrate dataset distribution.

To enable the evaluation of Just-in-Time Responsiveness and eliminate ambiguity in answer intervals, human annotators are required to mark clear time interval boundaries based on the completeness of objects within frames or the start/end of events. Simultaneously, questions with ambiguous references are filtered out (e.g., "Remind me the location of the ceramic bowl." where multiple ceramic bowls might be present in different locations). Each sample's question, answer, and corresponding answer interval are verified by two annotators. This rigorous verification process resulted in a dataset of 2264 verified question-answer instances. Notably, every answer in the dataset is associated with precise temporal annotations. Statistical information regarding the annotated data is presented in Fig. 3.

## 2.2 Evaluation Metric in ESTP

To comprehensively measure performance along three key evaluation aspects – answer quality, response timing, and temporal precision – we introduce the ESTP-F1 score. Here, we denote a ground truth item as $g_k$ with content $o_k$, and a prediction as $\hat{p}_l$ with content $\hat{o}_l$ and time $\hat{t}_l$. Evaluation components are defined for matched pairs $(\hat{p}_l, g_k)$, where $\hat{p}_l$ is a prediction that temporally matches $g_k$. For answer quality, an LLM [10] is used to measure correctness, defined as a score $\mathcal{S}_{\text{answer}}(\hat{o}_l, o_k)$ for the predicted content $\hat{o}_l$ relative to the ground truth content $o_k$. For evaluating response timing, we go beyond simply considering recall (which inherently accounts for False Negatives (FN)) and employ a score $\mathcal{S}_{\text{time}}(\hat{t}_l, g_k)$ to more precisely measure timeliness. Furthermore, for temporal precision, we introduce precision, utilizing False Positives (FP) as a penalty term. These components contribute to the aggregated ground truth score $S(g_k)$, which replaces the traditional binary TP count. The ESTP-F1 score is computed as:

$$\text{ESTP-F1} = \frac{2 \times \sum_{k=1}^{M} S(g_k)}{2 \sum_{k=1}^{M} S(g_k) + \mathsf{FP} + \mathsf{FN}}, \tag{1}$$

where $M$ is number of GT. High answer quality (reflected by a high $S_{\text{answer}}$ score), effective response timeliness (characterized by high $S_{\text{time}}$ for on-time responses and a low False Negative (FN) rate), and high precision (indicated by a low False Positive (FP) rate) collectively contribute to a high ESTP-F1 score. More details are provided in the Appendix.

# 3 Methodology: VideoLLM-EyeWO

In this section, we introduce a technical pipeline designed for the ESTP task. For the data engine, utilizing the Ego4D [17] training set and a three-stage generation pipeline as introduced in Sec. 1, we generate 60K single-turn and 20K multi-turn questions, as shown in Fig. 4. Each generated instance includes questions, answers, and their corresponding valid answer intervals (named as ESTP-IT). See Appendix for data engine details. Subsequently, we detail the problem definition and preliminary, the multi-stage training strategies, and the proactive dynamic compression technique in respective subsections.

## 3.1 Problem Definition and Preliminary

**Problem Definition.** Given a streaming video input and a sequence of emerging queries $\mathcal{Q} = \{(q_i, t_{q_i})\}$, where $q_i$ is the query content and $t_{q_i}$ is the query timestamp. At each timestep $t$ following a query (i.e., $t > t_{q_i}$), the model must leverage its historical memory $H_t$ (including visual input history and past query-response interactions), while concurrent observation $O_t$, to decide whether to perform a response action and generate corresponding content. The model's decision-making process at time $t$ can be formulated as selecting the optimal action $A_t$ from a predefined set $A$:

$$A_t = \text{argmax}_{a \in A} P_\theta(A_t = a \mid q_i, O_t, H_t). \tag{2}$$

Here, $\theta$ represent model parameter, $A_t$ is the model's action at time $t$, and $A$ is the set of possible actions. Notably, while previous work typically considers an action space that only includes $a_{\text{silence}}$ (staying silent) and $a_{\text{response}}$ (executing a response and generating a reply), we expands this by including the action $a_{\text{ask\_high}}$ (requesting a high-resolution frame), as introduced in Sec. 3.2 Stage-1.

**Preliminary.** LIVE [5] utilizes ground truth containing timestamps and applies cross-entropy supervision [48] to the model's action output at each timestep. Specifically, if the current time $t$ falls within a ground truth response region (denoted as $t \in \mathcal{T}_{\text{timestamp}}$), the model is supervised to execute the response action ($a_{\text{response}}$) and generate a reply, incorporating a language modeling loss $\mathcal{L}_{\text{LM}}$ [59, 11, 48]. Otherwise, it is supervised to remain silent ($a_{\text{continue}}$). This is formulated as:

$$\mathcal{L}(t) = \begin{cases} -\log P_\theta(a_{\text{response}} \mid q_i, O_t, H_t) + \omega \mathcal{L}_{\text{LM}}(t) & \text{if } t \in \mathcal{T}_{\text{timestamp}}, \\ -\log P_\theta(a_{\text{continue}} \mid q_i, O_t, H_t) & \text{otherwise}, \end{cases} \tag{3}$$

where, $\omega$ is a balancing coefficient weighting the language modeling objective.

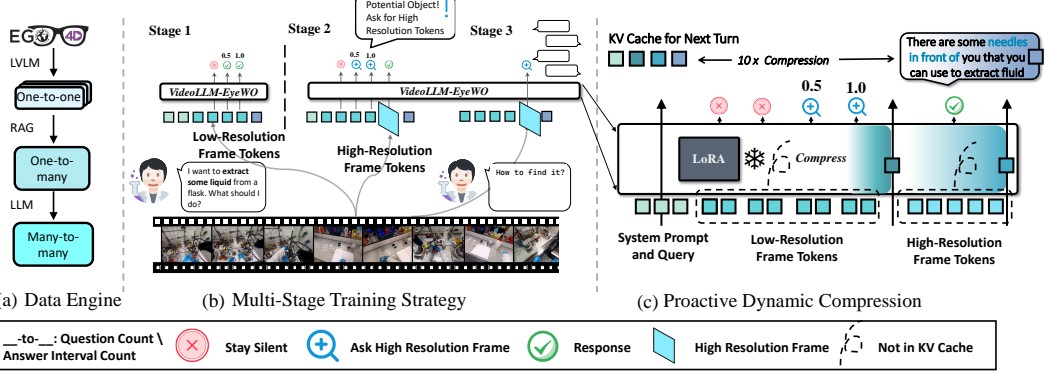

Figure 4: Overview of the proposed pipeline. The figure illustrates the three main components: (a) **Data Engine** (ESTP-Gen), which automatically generates diverse, multi-turn QA data through a three-stage pipeline. (b) **Multi-Stage Training Strategy** incrementally builds the model from basic responsiveness to proactive just-in-time accuracy, and ultimately to achieving multi-turn coherence, detailed in Section 3.2. (c) **Proactive Dynamic Compression** detailed in Section 3.3.

## 3.2 Multi-Stage Training Strategy

Following [5], VideoLLM-EyeWO utilizes the same network architecture and is trained using LoRA [20]. However, the single-stage training and simple binary supervision strategy employed in [5] can lead to training conflict due to the high similarity of adjacent frames in streaming inputs. This conflict necessitates a difficult trade-off between over-extended and under-responsive. To address these limitations, we employ a multi-stage training strategy designed to progressively endow the model with response capabilities. The following subsections detail each stage of this training strategy.

**Stage-1 : Passive Interval Responsivenes.** To provide the basic ability for autonomous response triggering, we leverage the valid answer intervals within the ESTP-IT to achieve a progressive transition from silence to response. Specifically, if current time $t$ falling within a valid answer interval (where $\mathcal{T}_{\text{interval}}$ is defined as the set of all such intervals $[s_i, e_i]$), we apply a weighted degree of response supervision, rather than direct binary classification, using the following loss function:

$$\mathcal{L}(t) = \begin{cases} -\log\left(f\left(\frac{|t-e|}{|s-e|}\right) \cdot P_\theta(a_{\text{response}} \mid q_i, O_t, H_t)\right) + \omega\mathcal{L}_{\text{LM}}(t) & \text{if } \boxed{\exists[s,e] \in \mathcal{T}_{\text{interval}}, t \in [s,e]} \\ -\log P_\theta(a_{\text{continue}} \mid q_i, O_t, H_t) & \text{otherwise} \end{cases},$$

(4)

The function $f$ is a linear decrease map used as a weighting factor applied to the response probability loss. The highlight in Equ. 4 is used to distinguish the components specific to this stage.

**Stage-2 : Proactive just-in-time responsiveness and accurate answering.** To use fine-grained visual details to pinpoint both the correct response moment and the accurate answer, we train the model to actively request high-resolution frames during uncertain timestamps in this stage. Specifically, we first introduce a third predefined action $a_{\text{ask\_high}}$. When the model executes this action at time $t$, it triggers the acquisition of the high-resolution frame $O_t^h$ corresponding to the current observation $O_t$ using the following loss function for training: $\mathcal{L}_{\text{ask\_high}}(t)$:

$$\mathcal{L}_{\text{ask\_high}}(t) = \begin{cases} -\log\left(f\left(\frac{|t-e|}{|s-e|}\right) \cdot P_\theta(\boxed{a_{\text{ask\_high}}} \mid q_i, O_t, H_t)\right) & \text{if } \boxed{t \in \mathcal{T}_{\text{uncertain}}} \\ -\log P_\theta(a_{\text{continue}} \mid q_i, O_t, H_t) & \text{otherwise,} \end{cases},$$

(5)

where $\mathcal{T}_{\text{uncertain}}$ denotes the set of the model's uncertain (see more detail in Appendix D Stage-2 Input). We use this loss to enable the model to acquire the ability to request high-resolution frames, and then based on the more detailed information, determine whether it is the correct time to respond and provide a more accurate answer, using the following loss:

$$\mathcal{L}_{\text{determine}}(t) = \begin{cases} -\log P_\theta(a_{\text{response}} \mid q_i, O_t, H_t, \boxed{O_t^h}) + \omega\mathcal{L}_{\text{LM}}(t) & \text{if } t \in \mathcal{T}_{\text{timestamp}} \\ -\log P_\theta(a_{\text{continue}} \mid q_i, O_t, H_t, \boxed{O_t^h}) & \text{otherwise} \end{cases},$$

(6)

where $O_t^h$ represents the high-resolution frame acquired at time $t$. The overall loss function at timestep $t$ is the sum of the two components:

$$\mathcal{L}(t) = \mathcal{L}_{\text{ask\_high}}(t) + \mathcal{L}_{\text{determine}}(t) \tag{7}$$

See appendix for detailed uncertain timestamps $\mathcal{T}_{\text{uncertain}}$ identified.

**Stage-3 : Coherence across multi-turn QA.** Building upon the model's acquired proactive and timely response capabilities, we introduce a separate training stage. Specifically, this stage involves training solely on multi-turn question, with the aim of further improving its contextual understanding while preserving its timely responsiveness.

### 3.3 Proactive Dynamic Compression Mechanism

In order to ensure memory efficiency as the number of incoming frames grows over time, we propose the Proactive Dynamic Compression Mechanism, which applies two levels of token compression and employs a uniform compression method, detailed respectively in the following two subsections.

**Two-Level Compression.** In contrast to fixed compression rates [29, 34, 4] and steps [43, 39, 58, 63], our mechanism leverages the streaming nature to allow the model to proactively determine both when to compress and which compression level to apply. Regarding the timing of compression, after the model generates a response, the preceding visual input and the response content itself form a natural segment or processing unit. Simultaneously, lower compression rates are applied to question-relevant content such as high-resolution frames, while higher rates are applied to other content, with these decisions proactively made by the model. Specifically, after a response, a fixed number of compression tokens (e.g., 1) are used to compress the preceding content, absorbing information from potentially many low-resolution frames or a single high-resolution frame. This approach naturally achieves a high compression rate for redundant parts of the past content, resulting in an average token usage of only about one-tenth of the original sequence.

**Uniform Compression Method.** For achieving two-level compression, we employ a Uniform Compression Method. Specifically, unlike methods using additional compression modules [34, 33], we insert a special compression token ($\langle\text{ct}\rangle$) after segments of original input, namely after single high-resolution frames, after multiple low-resolution frames, and after answer. This token is initialized using the text embedding of the "<EOS>" token. Leveraging the properties of the causal self-attention mechanism, this token prompts the LLM to compress the information from the preceding segment into a compact representation stored in the KV cache.

During training, inspired by [43], the LLM is trained to process response turns sequentially. A response turn refers to a turn of interaction, typically a comprising visual input and a model's response. Training for the Proactive Dynamic Compression Mechanism, including the integration of high-resolution frame requests, commences in Stage 2 of our multi-stage training strategy to ensure manageable training memory overhead.

## 4 Experiment

### 4.1 Baseline and Evaluation Settings

We evaluate three categories of models in this study: Offline MLLMs, VLMs, and Online MLLMs.

For **Offline MLLMs** we selected representative models from different open-source MLLM families, including LLaVA-OneVision [23], Qwen2-VL [49], MiniCPM-V [56], LLaVA-NeXT-Video [24], and InternVL-V2 [7]. As offline MLLMs lack inherent proactive response capability, following previous studies [27, 26, 51, 5], we employed two evaluation settings: (1) *Response-in-Last*: The model processes the complete video and is tasked with generating textual reply with timestamps. (2) *Polling Strategy*: The model is periodically queried at fixed time intervals. If the model indicates readiness, it is then prompted to generate the answer. Specific details regarding the prompts and hyperparameter used in these settings are provided in Appendix.

Regarding **VLMs**, following the approach of SDQES [13], we selected CLIP [36], LaViLa [64], and EgoVLP [28] for evaluation. These models were evaluated by computing the similarity between each frame and the query, using $0.5$ as the threshold to determine responsiveness. Notably, as these models cannot generate open-ended replies, their reply score is set to $0$.

| Model | Explicit Proactive Task | | | | | | | | | Implicit Proactive Task | | | | | Contextual Q | | | Overall |
|---|---|---|---|---|---|---|---|---|---|---|---|---|---|---|---|---|---|---|
| | OR | AP | TRU | OL | OSC | EOL | EOSC | AR | All | OFR | IFR | NAR | TU | All | ORC | TRC | All | |
| *Offline MLLMs Response-in-Last* | | | | | | | | | | | | | | | | | | |
| LLaVA-OneVision | 7.2 | 11.5 | 4.9 | 10.0 | 4.9 | 6.9 | 5.6 | 3.2 | 6.8 | 3.8 | 6.3 | 11.6 | 29.8 | 12.9 | 10.8 | 5.7 | 8.2 | 8.7 |
| Qwen2-VL | 11.7 | 8.1 | 14.9 | 10.5 | 1.7 | 8.9 | 10.6 | 6.0 | 9.0 | 10.2 | 4.4 | 26.5 | 49.5 | 22.6 | 13.3 | 9.4 | 11.3 | 13.3 |
| MiniCPM-V | 12.3 | 12.6 | 10.7 | 13.7 | 8.6 | 7.5 | 11.9 | 5.5 | 10.4 | 11.8 | 9.2 | 36.0 | 55.3 | 28.1 | 32.6 | 25.4 | 29.0 | 18.1 |
| LLaVA-NeXT-Video | 8.3 | 9.4 | 7.4 | 10.2 | 7.8 | 7.4 | 10.3 | 5.6 | 8.3 | 6.4 | 6.7 | 21.1 | 45.9 | 20.0 | 10.1 | 9.8 | 9.9 | 11.9 |
| InternVL-V2 | 9.3 | 14.6 | 9.5 | 10.6 | 1.7 | 6.3 | 3.0 | 3.6 | 7.3 | 3.3 | 9.2 | 15.5 | 28.2 | 14.0 | 16.9 | 15.6 | 16.2 | 10.5 |
| *VLMs for Streaming Detection* | | | | | | | | | | | | | | | | | | |
| CLIP | 7.3 | 9.5 | 7.4 | 8.5 | 1.8 | 4.7 | 2.2 | 2.7 | 5.5 | 2.8 | 5.2 | 51.3 | 29.3 | 22.2 | 4.6 | 3.8 | 4.2 | 10.1 |
| LaViLa | 8.4 | 10.7 | 9.0 | 9.1 | 3.1 | 5.4 | 3.6 | 4.3 | 6.7 | 7.8 | 10.0 | 56.2 | 34.4 | 27.1 | 9.4 | 28.9 | 19.2 | 14.3 |
| EgoVLP | 10.5 | 11.0 | 8.7 | 8.5 | 5.5 | 5.6 | 5.3 | 4.4 | 7.4 | 6.2 | 10.7 | 58.4 | 48.3 | 30.9 | 8.0 | 25.3 | 16.6 | 15.5 |
| *Offline MLLMs Polling Strategy* | | | | | | | | | | | | | | | | | | |
| LLaVA-OneVision | 8.3 | 8.8 | 22.8 | 25.4 | 13.5 | 9.8 | 9.6 | 10.3 | 13.6 | 20.3 | 20.9 | 35.9 | 49.9 | 31.8 | 14.6 | 1.9 | 8.2 | 18.0 |
| Qwen2-VL | 13.7 | 13.5 | 15.4 | 29.5 | 8.0 | 15.4 | 16.6 | 10.9 | 15.4 | 17.8 | 19.8 | 56.4 | 63.1 | 39.3 | 13.0 | 7.7 | 10.4 | 21.3 |
| MiniCPM-V | 14.9 | 16.8 | 17.1 | 26.8 | 7.7 | 12.9 | 12.5 | 13.1 | 15.2 | 15.9 | 21.0 | 46.8 | 62.2 | 36.5 | 24.3 | 28.9 | 26.6 | 22.9 |
| LLaVA-NeXT-Video | 15.6 | 14.6 | 21.9 | 26.8 | 12.8 | 14.2 | 13.5 | 12.3 | 16.5 | 18.6 | 23.2 | 44.9 | 51.6 | 34.6 | 19.9 | 7.7 | 13.8 | 21.3 |
| InternVL-V2 | 11.3 | 5.9 | 7.0 | 10.1 | 0.7 | 2.7 | 5.2 | 2.2 | 5.6 | 8.3 | 2.9 | 4.3 | 11.2 | 6.7 | 6.1 | 5.3 | 5.7 | 5.9 |
| *Online MLLMs* | | | | | | | | | | | | | | | | | | |
| LIVE(threshold=0.8) | 9.7 | 11.0 | 7.4 | 10.8 | 1.9 | 6.0 | 3.6 | 5.6 | 7.0 | 4.2 | 7.4 | 12.9 | 12.8 | 9.3 | 19.6 | 13.8 | 11.8 | 9.1 |
| LIVE(threshold=0.9) | 11.2 | 13.9 | 7.9 | 13.2 | 5.6 | 9.4 | 6.0 | 8.9 | 9.5 | 5.8 | 8.9 | 41.0 | 46.7 | 25.6 | 11.3 | 26.5 | 18.9 | 15.5 |
| MMDuet | 7.2 | 10.3 | 17.6 | 10.2 | 4.2 | 6.1 | 8.8 | 8.5 | 9.1 | 10.0 | 7.7 | 50.1 | 69.1 | 34.2 | 17.4 | 23.1 | 20.3 | 17.8 |
| **VideoLLM-EyeWO(Ours)** | 26.6 | 26.6 | 25.1 | 26.8 | 19.8 | 22.3 | 20.8 | 20.7 | 23.6 | 24.8 | 31.0 | 75.3 | 78.7 | 52.5 | 39.5 | 47.8 | 43.6 | 34.7 |

Table 2: Experimental results of various models evaluated on the ESTP-Bench. We present performance across Explicit Proactive, Implicit Proactive, and Contextual Question task types, as well as the Overall score, for Offline MLLMs (Response-in-Last and Polling Strategy), VLMs for streaming detection, and Online MLLMs. Deep blue highlights the best overall performance, while blue indicates the best performance within each model category and evaluation setting group.

For **Online MLLMs**, we selected VideoLLM-Online [5] and MMDuet [51], which provide open-source weights and streaming inference code, for evaluation. For VideoLLM-Online, we experimented with different thresholds to assess its performance variations.

## 4.2 Benchmarking in ESTP-Bench

**Comparative Analysis of Baseline Models.** Tab. 2 shows the performance of different models across three proactive types and fourteen task types under various evaluation settings, the experimental results consistently demonstrate that ESTP tasks pose significant challenges for all current types of models. Analysis revealed variations across model categories, with certain models exhibiting stronger capabilities within their respective groups (e.g., MiniCPM-V [56] and QwenVL-2 [49] among offline MLLMs aligning with previous work [8], and temporal VLMs like LaViLa [64] and EgoVLP [28] outperforming spatially-focused models like CLIP [36]). Furthermore, the evaluation strategy significantly impacts performance. Specifically, offline MLLMs showed a notable disparity, performing on average better under the Polling Strategy compared to the Response-in-Last strategy, with improvements up to 5.4%. This highlights the effectiveness of ESTP-Bench in evaluating models from a timeliness perspective and underscores the limitations in temporal grounding of existing offline models.

**Performance of VideoLLM-EyeWO Against Baselines.** As presented in Tab. 2, our proposed model achieved significant performance improvements across all proactive tasks. Compared to the baseline videoLLM-Online [5], our model demonstrated an improvement of +19.2%. Furthermore, it outperformed the best-performing model, MiniCPM-V [56](using the Polling strategy), by +11.8%.

## 4.3 In-Depth Analyses in ESTP-Bench

**Challenges with Coherent and Contextual Questions:** Fig. 5 illustrates the average performance of different models across 14 tasks. (NAR) and (TU) exhibit significantly higher performance compared to other tasks. Upon visualizing the proportion of valid answer intervals relative to the input video duration for these two tasks, we observe that this proportion is substantially higher than for other tasks. This is attributed to these annotations originating from the raw GoalStep [44] labels, which involve segmenting continuous actions towards a consistent goal, thereby leading to a larger proportion of valid answer interval within the video and consequently, higher Recall. Conversely, for the (TRC) task, which also derives from the same original annotations and possesses a high proportion of valid answer interval, both Recall and overall performance significantly decrease. This marked performance drop underscores the significant challenge that proactive coherence and understanding contextual information pose for existing models.

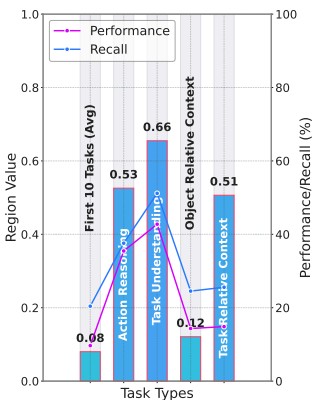 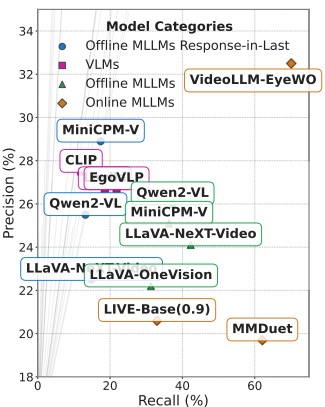 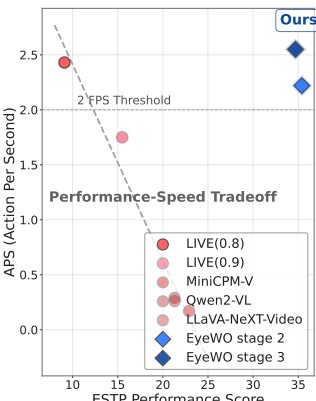

Figure 5: Average performance and Ground Truth interval proportion across 14 tasks, illustrating challenges with coherent and contextual questions.

Figure 6: Recall-Precision trade-off for different models and evaluation settings, highlighting the difficulty in responding only when necessary.

Figure 7: Action Per Second versus ESTP Score for various models, measured on an A40 GPU, demonstrating synchronization efficiency challenges.

**Difficulty in Responding Only When Necessary:** Fig. 6 presents the relationship between recall and precision for different models under various evaluation settings. We observe a prevalent negative correlation between recall and precision among most models. For instance, MMDuet achieves exceptionally high recall but at the expense of low precision. This trade-off indicates the struggle of existing models to provide proactive yet precise responses.

**Synchronization Efficiency Challenges:** Fig. 7 illustrates the inherent performance-speed tradeoff in ESTP tasks by plotting Action Per Second (APS) against Performance Score for various models. Existing methods often lie along a clear tradeoff curve, where higher performance is typically associated with lower APS, highlighting the difficulty in achieving both simultaneously. As seen for offline MLLMs using the Polling strategy, achieving high performance while maintaining sufficient speed for real-time synchronization remains challenging. Even approaches near the input frame rate (e.g., LIVE at ~2 FPS) may demonstrate suboptimal performance. This underscores the significant challenges current models face in achieving both high task performance and effective synchronization with the dynamic video stream.

### 4.4 Evalutation of VideoLLM-EyeWO

**Evaluating Zero-Shot Capability in Online/Offline Tasks.** Table 3 presents a performance comparison of our model against the baseline on both online and offline tasks. We selected VideoLLM-online [5] as our baseline, given that it shares the same base model (LLaMA3 [16] and SigLIP [60]) and data source (Ego4D) as our own mode. For the online task, we utilize OvO-Bench [26] as a recognized benchmark. For the offline task, following [12], we evaluate our model on the multiple-choice subset of the QAEGO4D-test benchmark [3]. The 'Online' setting involves posing questions as soon as the relevant answer segment appears, whereas the 'Offline' setting involves questioning after the entire video has been presented. The experimental results demonstrate the generalization capability of our model across these distinct tasks.

| Model | Online Task: OVO-Bench | | | | | | | | | | | Offline Task |
| | Real-Time Perception | | | | | | | Backward Tracing | | | | QAEGO4D_MC |
| | OCR | ACR | ATR | STU | FPD | OJR | Avg. | EPM | ASI | HLD | Avg. | Online | Offline |
| VideoLLM-online | 8.05 | 23.85 | 12.07 | 14.04 | **45.54** | 21.20 | 20.79 | 22.22 | 18.80 | **12.18** | 17.73 | 29.80 | 30.20 |
| Ours (VideoLLM-EyeWO) | **24.16** | **27.52** | **31.89** | **32.58** | 44.55 | **35.87** | **32.76** | **39.06** | **38.51** | 6.45 | **28.00** | **36.20** | **33.00** |

Table 3: Detailed Performance Evaluation on OVO-Bench [26] and QAEGO4D [3] Tasks.

**Evaluating Architecture Generalizability on Offline Tasks** As presented in Tab. 4, our model demonstrated comprehensive performance improvements on five tasks related to traditional temporal summarization and forecasting problems. The performance gain reached up to +2.8%, which indicates that our proposed model architecture can effectively generalize to other offline tasks.

| Method | COIN Benchmark | | | | |
|---|---|---|---|---|---|
| | Step | Task | Next | Proc | Proc+ |
| ClipBERT [22] | 30.8 | 65.4 | - | - | - |
| VideoLLM-online-7B-v1 [5] | 59.8 | 92.1 | 44.7 | 47.9 | 52.9 |
| VideoLLM-online-8B-v1+ [5] | 63.1 | 92.6 | 49.0 | 49.7 | 53.6 |
| VideoLLM-MOD [52] | 63.4 | 92.7 | 49.8 | 49.8 | 53.3 |
| Ours (LLaMa3 [16, 47]) | 65.9 | 92.7 | 50.9 | 50.8 | 54.7 |
| Ours (LLaMa3.1 [16, 47]) | **66.0** | **93.3** | **51.5** | **51.1** | **55.5** |

Table 4: COIN [46] Benchmark Top-1 Accuracy comparison across different methods.

## 5 Related Work

**Streaming Video Understanding.** Unlike traditional offline video understanding tasks [14, 66, 1, 50] which only allow for question answering after processing the entire video, Online Video Understanding tasks aim to evaluate the ability to respond to questions that arise at any time, based on past information and current observations from a sequential video stream input. Previous work [61, 26, 27] has employed various question types presented during the video stream, such as object or event perception, leveraging different spatiotemporal cues to comprehensively evaluate streaming video understanding capabilities. However, for proactive tasks where questions often require reasoning [65] beyond the current frame, existing methods often exhibit limited question diversity and lack contextual continuity across queries. Furthermore, evaluation in previous work has often overlooked efficiency and timeliness.

To specifically endow MLLMs [42] with proactive response capability, VideoLLM-Online [5] proposed the LIVE training framework, which supervises the model's output at each frame. Subsequent work has focused on improving training efficiency [52], inference speed [33], and response capability [51]. However, these methods often generate under-responsive or over-extended answers and struggle to adapt to continuous, multi-turn conversation.

**Egocentric Video Understanding.** Compared to traditional third-person video understanding, egocentric video data poses unique challenges, such as the scarcity of large-scale annotated datasets and the inherent narrow and often unstable viewpoint. However, recent work [17, 18, 9, 44] has contributed massive egocentric video datasets and fundamental annotations, significantly benefiting the community. Prior work has typically focused on classic egocentric video understanding tasks such as activity recognition [25, 38], temporal grounding [55, 31, 41], and hand-object detection [2, 21], which are often limited by a closed vocabulary. Other efforts have focused on offline video understanding tasks, such as QaEgo4D [3], EgoSchema [30], and MM-Ego [57]. While SDQES [13] introduced a streaming detection benchmark for egocentric video, it primarily evaluates similarity-based responsiveness and lacks the scope to assess the open-ended generative and conversational capabilities required from Online MLLMs in proactive scenarios.

## 6 Conclusion

We definite an novel AI assistant's task of proactive, synchronized question answering from ego-streaming video, targeting the key properties of proactive coherence, just-in-time responsiveness, and synchronized efficiency. Our contributions—the ESTP-Bench with its ESTP-F1 metric for evaluation, and a novel technical pipeline incorporating a data engine, multi-stage training, and proactive dynamic compression—enable our model to effectively tackle these properties. This approach outperforms multiple baselines across diverse online and offline benchmarks.

**Acknowledgement:** This work is supported in part by the National Natural Science Foundation of China under Grant No.62206174 and No.62576365.

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

# A  Ablation Study of VideoLLM-EyeWO

| Method | Single Question | | Contextual Question | |
|---|---|---|---|---|
| | Performance ↑ | KV Cache Size ↓ | Performance ↑ | KV Cache Size ↓ |
| LIVE | 14.9 | 9636.0 | 18.9 | 31199.5 |
| + ESTP-IT | 22.0 | 7859.1 | 25.7 | 28236.4 |
| Stage-0 | 24.9 | 7988.2 | 23.0 | 17567.6 |
| *with increased proactive dynamic compression mechanism* | | | | |
| + Stage-1 ask high frame | **34.0** | 1182.8 | 38.7 | 3731.8 |
| + Stage-2 | 33.2 | 942.0 | **43.6** | 3242.8 |

Table 5: Ablation study results on ESTP bench

Tab. 5 details the results of our ablation study on the ESTP benchmark:

1. (+*ESTP-IT*) enhanced the LIVE baseline's performance on both Single and Contextual Question tasks, increasing it by +7.1 and +6.8 respectively, **thereby demonstrating the effectiveness of ESTP-IT**.
2. (*Stage-0*) addressed the training conflicts stemming from simple binary supervision, enabling performance improvements without requiring any manual threshold tuning, **which demonstrates the model's acquisition of a basic ability to trigger responses**.
3. With the increased proactive dynamic compression mechanism, the model's KV cache consumption was significantly reduced, **requiring on average only about 0.11% of the baseline**.
4. (+*Stage-1*) significantly boosted Single Question performance to 34.0 and Contextual Question performance jumped to 38.7 by **incorporating the mechanism for actively requesting high-resolution frames for scrutiny alongside initial compression**.
5. (+*Stage-2*) **further improved contextual coherence and refined compression**, enabling the model to achieve a gain of +4.9 on Contextual tasks, reaching 43.6. Simultaneously, the more accurate and efficient responses further reduced memory consumption to minimal levels.

# B  ESTP-Bench: Dataset and Evaluation

In this section, we first introduce the detailed description for each task within ESTP-Bench. Subsequently, we present the annotation tool and pipeline employed for these tasks. Finally, we provide a detailed description of the ESTP-F1 metric.

## B.1  Task Explanation

**Explicit Proactive Tasks** require detecting and responding to queries that directly reference visual cues. This category includes eight distinct task types:

1. [OR] **Object Recognition**: Detect and identify specific objects.

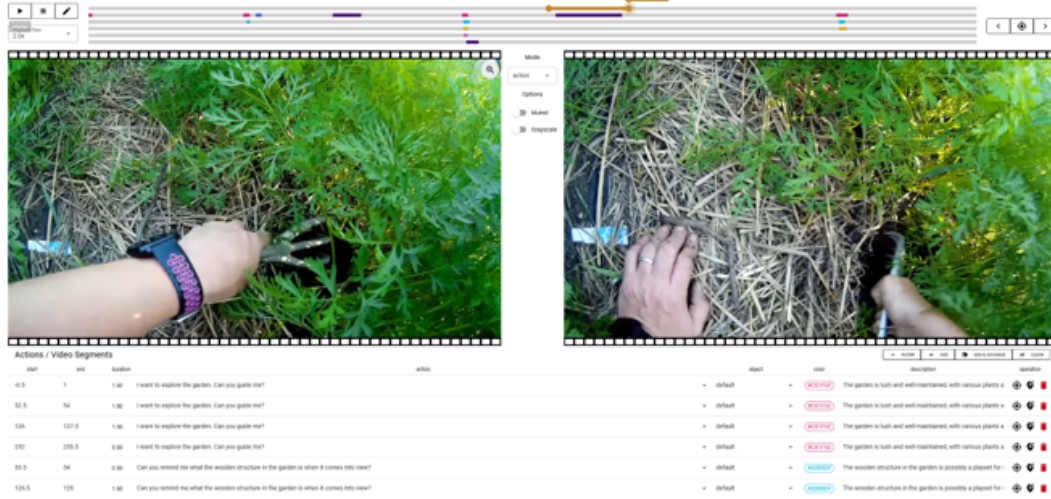

Figure 8: **Interface of the annotation tool.** This tool enables annotators to define valid answer intervals on the video timeline, author questions, and input corresponding answers. For clarity, multiple answers and their associated intervals for a single question are visually linked by a shared color.

2. [AP] **Attribute Perception**: Recognize object attributes.
3. [TRU] **Text-Rich Understanding**: Interpret and explain text content.
4. [OL] **Object Localization**: Identify spatial relations between objects.
5. [OSC] **Object State Change**: Detect object state transitions.
6. [EOL] **Ego Object Localization**: Localize objects relative to the ego.
7. [EOSC] **Ego Object State Change**: Track object state changes from the ego's view.
8. [AR] **Action Recognition**: Detect and classify specific actions.

**Implicit Proactive Tasks** require performing inference and scene understanding beyond direct observation. Task types include:

1. [OFR] **Object Function Reasoning**: Identify functional objects and guide their use.
2. [IFR] **Information Function Reasoning**: Extract and provide information from source objects.
3. [NAR] **Next Action Reasoning**: Predict and recommend next user actions.
4. [TU] **Task Understanding**: Comprehend user goals and offer task guidance.

**Contextual Proactive Tasks** require maintaining dialogue history and visual coherence over time. Task types include:

1. [ORC] **Object Relative Context**: Track an object's attributes, position changes, and function across question sequences.
2. [TRC] **Task Relative Context**: Monitor user task progress, action correctness, and provide sequential guidance.

See Sec. G for data examples.

## B.2 Annotation Tool and Pipeline

**Annotation Tool.** As shown in Fig. 8, our annotation tool [62] facilitates the labeling process: annotators can adjust the start and end timestamps of valid answer intervals in the upper region, while authoring questions in the bottom-left area and completing answers in the corresponding region on the right. Notably, a single question can be associated with multiple answers and their corresponding intervals, which are visually linked through a shared color for clarity.

**Annotation Construction Pipeline.** As a first step, we leverage LLMs [10] to convert raw annotations into initial question-answer pairs. For Action Recognition (AR) tasks, we utilize Ego4D [17]

basic Narration raw annotations, which comprise descriptions of action-related events and their corresponding timestamps. For Next Action Reasoning (NAR), Task Understanding (TU), and Task Relative Context (TRC) questions, we leverage Ego4D GoalStep [44] annotations, which provide a task goal, multiple task steps, and their corresponding temporal intervals. For these aforementioned tasks, we directly prompt LLMs for question generation. For all other task types, question generation is performed through the data engine.

Subsequently, the data undergoes a human annotation phase. For Action Recognition (AR), Next Action Reasoning (NAR), Task Understanding (TU), and Task Relative Context (TRC) tasks, given their origin from existing dataset [44], annotators primarily focus on three aspects: (1) detecting question ambiguity; (2) assessing the question-answer pair's matching quality and correcting erroneous wording; and (3) calibrating the answer interval timestamps. For all other task types, annotators are additionally required to generate questions and complete answers, inspired by the initial QA pairs, before executing the aforementioned refinement steps. Simultaneously, a double-check mechanism is implemented, where a second annotator independently re-executes the aforementioned three steps for verification.

### B.3 Evaluation Metric (ESTP-F1)

Let $\mathcal{G} = \{g_k = (o_k, [t_k^{\text{start}}, t_k^{\text{end}}], t_k^{\text{opt}})\}_{k=1}^M$ denote a set of $M$ ground truth items, where $o_k$ is the content, $[t_k^{\text{start}}, t_k^{\text{end}}]$ is the temporal interval, and $t_k^{\text{opt}} \in [t_k^{\text{start}}, t_k^{\text{end}}]$ is the optimal response timestamp. Let $\mathcal{P} = \{\hat{p}_l = (\hat{o}_l, \hat{t}_l)\}_{l=1}^N$ represent the set of $N$ predictions, where each prediction $\hat{p}_l$ consists of the predicted content $\hat{o}_l$ and a prediction timestamp $\hat{t}_l$. Following [13], we also introduce temporal tolerances: $\tau_{\text{ant}}$ (set to 1 second) for allowed anticipation before $t_k^{\text{start}}$, and $\tau_{\text{lat}}$ (set to 2 seconds) for allowed latency after $t_k^{\text{end}}$.

A prediction $\hat{p}_l$ is considered a **valid match** for a ground truth $g_k$ if its timestamp $\hat{t}_l$ falls within the interval $[t_k^{\text{start}} - \tau_{\text{ant}}, t_k^{\text{end}} + \tau_{\text{lat}}]$. We define $\mathcal{P}_k = \{\hat{p}_l \in \mathcal{P} \mid \hat{p}_l \text{ is a valid match for } g_k\}$ as the set of all such valid matching predictions for $g_k$.

For each prediction $\hat{p}_l$ that is a valid match for $g_k$, we compute a **match quality score** $\mathcal{S}(\hat{p}_l, g_k) \in [0, 1]$ to comprehensively evaluate its quality. This score is the average of a answer accuracy score and a timeliness score:

$$\mathcal{S}(\hat{p}_l, g_k) = \frac{\text{Score}_{\text{answer}}(\hat{o}_l, o_k) + \text{Score}_{\text{time}}(\hat{t}_l, g_k)}{10}. \tag{8}$$

The $\text{Score}_{\text{answer}}(\hat{o}_l, o_k) \in [1, 5]$ evaluates the semantic correctness of the predicted content $\hat{o}_l$ against the ground truth content $o_k$, typically assessed by a advanced Large Language Model [10].

The $\text{Score}_{\text{time}}(\hat{t}_l, g_k) \in [0, 5]$ assesses the temporal accuracy of the prediction timestamp $\hat{t}_l$ relative to the optimal response timestamp $t_k^{\text{opt}}$. The definition of $t_k^{\text{opt}}$ varies by task type:

- For tasks emphasizing immediate perception upon appearance (e.g., Object Recognition (OR), Attribute Perception (AP), Text-Rich Understanding (TRU), Object Localization (OL), Object Function Reasoning (OFR), Information Function Reasoning (IFR), and Object Relative Context (ORC)), $t_k^{\text{opt}}$ is set to $t_k^{\text{start}}$.

- For all other task types that may require more observation of event progression, $t_k^{\text{opt}}$ is the midpoint of the interval $[t_k^{\text{start}}, t_k^{\text{end}}]$, allowing sufficient time for information processing before a response.

The timeliness score is then calculated as:

$$\text{Score}_{\text{time}}(\hat{t}_l, g_k) = 5 - 5 \times \min\left(1, \frac{|\hat{t}_l - t_k^{\text{opt}}|}{\text{scale}_k}\right), \tag{9}$$

where $\text{scale}_k = \max(1, (t_k^{\text{end}} - t_k^{\text{start}}) + \tau_{\text{ant}} + \tau_{\text{lat}})$ is a normalization factor representing the total effective time window. This factor is used to prevent division by zero for point intervals (where $t_k^{\text{start}} = t_k^{\text{end}}$).

For each ground truth item $g_k$, an aggregated score $S(g_k) \in [0, 1]$ is determined by averaging the quality scores of its matching predictions:

$$S(g_k) = \begin{cases} \frac{1}{|\mathcal{P}_k|} \sum_{\hat{p}_l \in \mathcal{P}_k} \mathcal{S}(\hat{p}_l, g_k) & \text{if } \mathcal{P}_k \neq \emptyset \\ 0 & \text{if } \mathcal{P}_k = \emptyset \end{cases}. \tag{10}$$

This $S(g_k)$ provides a nuanced measure of performance for each ground truth item.

Finally, the **ESTP-F1 score** is calculated as:

$$\text{ESTP-F1} = \frac{2 \sum_{k=1}^{M} S(g_k)}{N + M - 2 \sum_{k=1}^{M} \mathbb{I}(S(g_k) \neq 0) + 2 \sum_{k=1}^{M} S(g_k)} \tag{11}$$

where $\mathbb{I}(\cdot)$ is the indicator function. This metric offers a comprehensive evaluation of a model's just-in-time proactive capability by jointly considering content accuracy and temporal alignment through the $S(g_k)$ scores. **In contrast to classic detection tasks, which may generate redundant proposals for the same region, streaming proactive tasks necessitate a single, well-considered judgment at each timestamp. Therefore, rather than selecting the best among multiple redundant proposals as a True Positive (TP), our evaluation metric calculates the average of all valid matches. This approach comprehensively assesses each of the model's actions, reflecting the unique temporal nature of streaming environments.**

## B.4 Data Cleaning and Filtering

To ensure the high quality and suitability of the egocentric video data for the ESTP tasks, a comprehensive data cleaning and filtering pipeline is applied, involving the following key steps:

1. Initial quality control was performed in accordance with [28] to identify and remove corrupted, incomplete, or visually ambiguous segments, ensuring the foundational integrity of the data.
2. Segments with a duration shorter than **250 seconds** were filtered out to ensure sufficient temporal context for learning proactive behaviors.
3. Segments where the narration annotation coverage fell below a threshold of **0.8** were discarded to ensure robust linguistic supervision and data richness.

This multi-stage filtering process results in a refined dataset optimized for the challenges of streaming proactive perception, ensuring both visual and linguistic fidelity.

## B.5 Data Engine: ESTP-Gen

As shown in Fig. 9, for automatically generates diverse, multi-turn data, we propose **ESTP-Gen**, a data engine that leverages VLMs [56, 49] and LLMs [10] to generate diverse proactive QA data from large-scale ego video datasets like Ego4D [17]. We apply a multi-stage pipeline including *Caption Generation*, *Key Segment Extraction*, and *Multi-Level QA Generation*. For simplification, in the main paper, we consolidate the Caption Generation and Key Segment Extraction steps into the one-to-one stage. **For videos spanning several minutes, directly generating QA for all segments proves challenging for entities that are persistently visible and static, as defining clear response boundaries for such entities is often ambiguous. Instead, we solely generate relevant QA from extracted key segments and complete answers by leveraging all available captions. This approach simultaneously ensures clear response boundaries and answer completeness.** The specific details of each stage below:

**Multi-Perspective Caption Generation:**

- **Narration:** Origin narration [44, 17] provides basic descriptions of human actions and their corresponding timestamps, serving as a foundational reference for our caption generation process.
- **Event Caption:** The original narration is typically structured as a simple subject-verb-object phrase, merely including basic category information for the actor, action, and manipulated object. It often lacks detailed information regarding object attributes, spatial-temporal location, state, or actions beyond the basic level. To generate more detailed captions specifically for events, we first follow the video clipping approach of EgoVLP [28] to obtain event-relevant video clips based

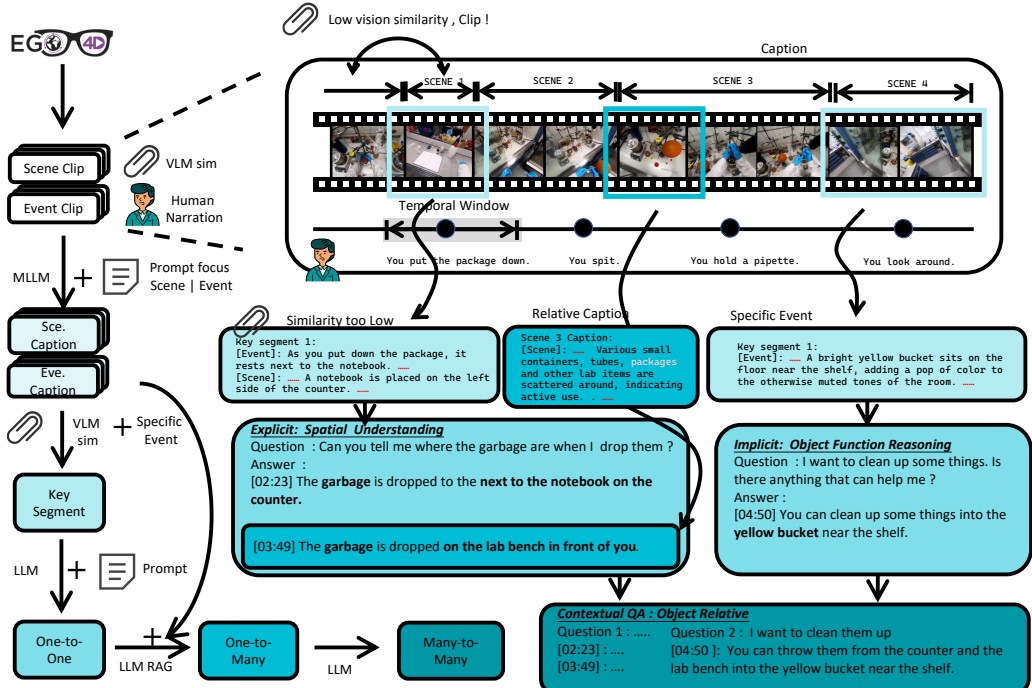

Figure 9: **ESTP-Gen Data Engine** is a multi-stage pipeline for automatically generating diverse, high-quality egocentric proactive QA data, which includes: (a) Multi-Perspective Caption Generation, creating Narration, Event, and Scene Captions to provide comprehensive foreground and background context. (b) Key Segment Extraction, identifying key segments characterized by significant visual changes and specific events. (c) **QA Generation Pipeline**, proceeding through three stages: One-to-One QA for initial pairs, One-to-Many Expansion, and Many-to-Many.

on the origin narration timestamps. Then, using lexical analysis [19], we extract the actions and interacted objects. Subsequently, we use an MLLM [56] to provide detailed descriptions of the action and the interacted object.

- **Scene Caption:** The Event Captions primarily focus on foreground directly involved in human interaction, potentially overlooking the broader background content within ego videos. Therefore, a detailed description of the video's background information is necessary to complement the Event Captions. First, we perform video scene segmentation. Then, for each video segment, we prompt an MLLM [56] to generate detailed descriptions of the background context. Specifically, we utilize SigLIP [60] to extract features from each video frame and compute the change from the previous frame using cosine similarity. Potential video scene boundaries are identified where the similarity is significantly below the video's average similarity. This segmentation method, based on visual content changes, preserves the comprehensive scene content within ego videos, facilitating effective description by the MLLM.

**Key Segment Extraction:** Avoiding questions about persistently visible visual and static content is a key challenge for data generation of proactive task. We address this by employing two criteria for key segment extraction aimed at identifying moments suitable for generating proactive QA: specific egocentric actions and visual similarity change. The key insight is that moments in ego videos characterized by significant visual changes and scene transitions are necessarily accompanied by specific egocentric events of the actor (e.g., "turn around," "walk," "step into"). In conjunction with visual similarity analysis, we extract visual segments exhibiting viewpoint changes to serve as the basis for QA generation. As illustrated in the portion of Fig. 9, we extract key segments for QA generation. These segments are identified by significant visual changes (e.g., "scene 1") and specific event (e.g., "You look around")

- **One-to-One QA:** Using diverse prompting strategies, we guide LLMs to generate questions, visual cues, and answers from captions. As illustrated in the portion of Fig. 9, the caption of discarding

a package prompted the LLM [10] to generate a question concerning the "location of dropped garbage", while the yellow bucket prompted the LLM [10] to generate a question regarding the "function of the item for helping clean".

- **One-to-Many Expansion:** We refine timestamps by aligning QA pairs with captions that fully contain the visual cues needed to answer, improving temporal coverage and QA consistency. As shown in the portion of Fig. 9, utilizing the LLM's Retrieval-Augmented Generation (RAG) capabilities, segments with similar visual cues are identified from all remaining captions to complete all related answers.

- **Many-to-Many Contextual QA:** Logically grouped questions from the same video are combined via LLMs to form coherent multi-turn contextual QA sets. As illustrated in the portion of the Fig. 9, the questions "location of dropped garbage" and "function of the item for helping clean" exhibit logical relevance, and are therefore merged into a coherent multi-turn contextual QA set.

## C   Model Input Example

In this section, we utilize input examples to more clearly illustrate the concepts presented in Sec. 3 of the main paper. We first present input examples for LIVE [5] (corresponding to Sec. 3.1 of the main paper), followed by input examples for our multi-stage training strategy (corresponding to Sec. 3.2 of the main paper). Finally, we will demonstrate Compression Token Insertion and related training details, further elucidating key aspects introduced in Sec. 3.3 of the main paper.

**LIVE Input.** An input example for the LIVE [5] model is provided below. Visual tokens [F] denote the tokens per frame. The number of visual tokens per frame, $|[F]|$, is set to 10, comprising 1 CLS token and $3 \times 3$ average-pooled spatial tokens for enhanced visual understanding. Tokens marked in purple indicate an output of $a_{\text{response}}$, while green tokens correspond to an output of $a_{\text{continue}}$. Text highlighted in red is subject to language model loss [59, 11, 48], whereas other textual elements receive no supervision. Some chat template strings have been omitted for better visualization.

---

**LIVE Example**

**[System Prompt]**
**Observation:** [F] [F] . . . **User:** Can you tell me where the garbage are when I drop them? **Observation:** [F][F]. . . [F][F]**Assistant:** The garbage is dropped to the next to the notebook on the counter. **Observation:** [F][F]. . . [F][F]**Assistant:** The garbage is dropped on the lab bench in front of you. **Observation:** [F] [F] . . . **User:** I want to clean them up. **Observation:** [F][F]. . . [F][F]**Assistant:** You can throw them from the counter and the lab bench into the yellow bucket near the shelf. **Observation:** [F] [F] . . .

---

**Stage-1 Input.** An input example for the Stage-1 training is provided below. The core distinction from the LIVE [5] example is the transformation of the abrupt supervision signal into a progressive one. Visually, as the purple color deepens, the weight of the supervision signal increases, indicating that the supervision signal becomes stronger as it approaches the optimal response timestamp.

---

**Stage-1 Example**

**[System Prompt]**
**Observation:** [F] [F] . . . **User:** Can you tell me where the garbage are when I drop them? **Observation:** [F][F]. . . [F]**Assistant:** The garbage is dropped to the next to the notebook on the counter. **Observation:** [F][F]. . . [F][F]**Assistant:** The garbage is dropped on the lab bench in front of you. **Observation:** [F] [F] . . . **User:** I want to clean them up. **Observation:** [F][F]. . . [F][F][F]**Assistant:** You can throw them from the counter and the lab bench into the yellow bucket near the shelf. **Observation:** [F] [F] . . .

---

**Stage-2 Input.** First, the set of uncertain timestamps $\mathcal{T}_{\text{uncertain}}$ (shown in Equ. 5) is automatically identified by perceiving timesteps where the model is likely to be confused or uncertain. Specifically, an initial training phase is conducted for one epoch using the $\mathcal{L}_{\text{ask\_high}}(t)$ loss, with $\mathcal{T}_{\text{uncertain}}$ initially set equal to $\mathcal{T}_{\text{interval}}$. An illustrative example is provided below. Compared to Stage 1, the $a_{\text{response}}$ is now replaced by $a_{\text{ask\_high}}$, with the blue tokens indicating receiving this new supervision.

> **[System Prompt]**
> **Observation:** [F] [F] ... **User:** Can you tell me where the garbage are when I drop
> them? **Observation:** [F][F]... [F][HF]**Assistant:** The garbage is dropped to the next to
> the notebook on the counter. **Observation:** [F][F]... [F][F][HF]**Assistant:** The garbage is
> dropped on the lab bench in front of you. **Observation:** [F] [F] ... **User:** I want to clean
> them up. **Observation:** [F][F]... [F][F][HF]**Assistant:** You can throw them from the
> counter and the lab bench into the yellow bucket near the shelf. **Observation:** [F] [F] ...

Subsequently, by comparing the model's predicted sequence against the ground truth sequence, we identify timestamps of erroneous $a_{\text{ask\_high}}$ predictions ($t_{\text{error}}$). For each such error timestamp $t_{\text{error}}$, we locate the nearest preceding timestamp $t_{\text{visual\_change\_prev}}$ characterized by a significant visual feature change, measured using visual [60] similarity. The set $\mathcal{T}_{\text{error}}$ is then defined as the union of intervals $[t_{\text{visual\_change\_prev}}, t_{\text{error}}]$, where for each response turn, $t_{\text{error}}$ corresponds to the erroneous prediction with the highest logit if multiple errors are present within that turn. An example is provided below, where red tokens indicate timesteps where the ground truth was $a_{\text{continue}}$ but the model erroneously predicted $a_{\text{ask\_high}}$.

> **[System Prompt]**
> **Observation:** [F] [F] ... **User:** Can you tell me where the garbage are when I drop them?
> **Observation:** [F][F]... [F][F]... [F][HF]**Assistant:** The garbage is dropped to the next to
> the notebook on the counter. **Observation:** [F][F]... [F][F]... [F][F][HF]**Assistant:** The
> garbage is dropped on the lab bench in front of you. **Observation:** [F] [F] ... **User:** I
> want to clean them up. **Observation:** [F][F]... [F][F]... [F][F][HF]**Assistant:** You can
> throw them from the counter and the lab bench into the yellow bucket near the shelf.
> **Observation:** [F] [F] ...

$\mathcal{T}_{\text{uncertain}}$ is then constructed by merging $\mathcal{T}_{\text{error}}$ with the Ground Truth relevant regions $\mathcal{T}_{\text{interval}}$. This allows the model to learn from mistakes and focus training on challenging moments, encouraging it to utilize scrutiny before making correct decisions in these uncertain scenarios. An illustrative example is provided below.

> **Stage-2 Example**
>
> **[System Prompt]**
> **Observation:** [F] [F] ... **User:** Can you tell me where the garbage are
> when I drop them? **Observation:** [F][F]... [F][F][F][HF]... [F][HF]**Assistant:** The
> garbage is dropped to the next to the notebook on the counter. **Observation:**
> [F][F]... [F] [F][F][HF]... [F][F][HF]**Assistant:** The garbage is dropped on the lab bench
> in front of you. **Observation:** [F] [F] ... **User:** I want to clean them up. **Observation:**
> [F][F]... [F][F]... [F][F][HF]**Assistant:** You can throw them from the counter and the
> lab bench into the yellow bucket near the shelf. **Observation:** [F] [F] ...

**Proactive Dynamic Compression Mechanism Example.** An example illustrating the insertion of compression tokens is provided below. Compared to the Stage-2 example, $\langle \text{ct} \rangle$ tokens are inserted after segments of input, specifically following multiple low-resolution frames, single high-resolution frames, and text replies.

> **Compression Example**
>
> **[System Prompt]**
> **Observation:** [F] [F] ... $\langle \text{ct} \rangle$ **User:** Can you tell me where the garbage are when I
> drop them? **Observation:** [F][F]... [F][F][F]$\langle \text{ct} \rangle$ [HF]$\langle \text{ct} \rangle$ ... [F]$\langle \text{ct} \rangle$ [HF]$\langle \text{ct} \rangle$ **Assistant:**
> The garbage is dropped to the next to the notebook on the counter. $\langle \text{ct} \rangle$ **Observation:**
> [F][F]... [F] [F][F]$\langle \text{ct} \rangle$ [HF]$\langle \text{ct} \rangle$ ... [F][F]$\langle \text{ct} \rangle$ [HF]$\langle \text{ct} \rangle$ **Assistant:** The garbage is dropped
> on the lab bench in front of you. $\langle \text{ct} \rangle$ **Observation:** [F] [F] ... $\langle \text{ct} \rangle$ **User:** I want to clean
> them up. **Observation:** [F][F]... [F][F]$\langle \text{ct} \rangle$ [HF]$\langle \text{ct} \rangle$ ... [F]$\langle \text{ct} \rangle$ [F][HF]$\langle \text{ct} \rangle$ **Assistant:**
> You can throw them from the counter and the lab bench into the yellow bucket near the
> shelf. $\langle \text{ct} \rangle$ **Observation:** [F] [F] ...

| | | Stage-1 | Stage-2 | Stage-3 |
|---|---|---|---|---|
| **Data** | Dataset | Ego4D: Narration Stream + GoalStep Stream ESTP-IT Single-Turn + Multi-Turn | ESTP-IT Single-Turn + Multi-Turn | ESTP-IT Multi-Turn |
| | #Samples | 113K + 21K 60K + 20K | 60K + 20K | 20K |
| **Model** | Base LLM | LLaMA3 | LLaMA3 | LLaMA3 |
| | Vision Encoder(s) | SigLIP | SigLIP | SigLIP |
| | Vision Token per Frame | 1+3x3 | Low Res: 1+3x3 High Res: 1+7x7 | Low Res: 1+3x3 High Res: 1+7x7 |
| | Connector | MLP | MLP | MLP |
| | Trainable | Connector(full) LLM(LoRA r=128, scale=256) | Connector(full) LLM(LoRA r=128, scale=256) | Connector(full) LLM(LoRA r=128, scale=256) |
| **Initialization** | Connector | N/A | Stage-1 Connector | Stage-2 Connector |
| | LLM | LLaMA3 | + Stage-1 LoRA | + Stage-2 LoRA |
| **Training** | Batch Size per Device | 1 | 1 | 1 |
| | Gradient Accumulation | 8 | 8 | 8 |
| | Learning Rate | 2e-4 | 1e-4 | 5e-5 |
| | Warm-up Ratio | 0.05 | 0.05 | 0.05 |
| | LR Scheduler | Cosine | Cosine | Cosine |
| | Optimizer | Adamw | Adamw | Adamw |
| | Epochs | 2 | 1 | 1 |
| | Precision | bf16 fp16 | bf16 fp16 | bf16 fp16 |

Table 6: Multi-Stage Training Plan in ESTP

During training, inspired by [43], the LLM is trained to process response turns sequentially. The initial sequence provided to the LLM is as follows:

> **[System Prompt]**
> **Observation:** [F] [F] ... $\langle ct \rangle$

Past visual tokens are compressed into a $\langle ct \rangle$. The System Prompt, containing essential system-level instructions [53], remains uncompressed. The subsequent sequence processed by the LLM is as follows, where gray portions indicate tokens stored in the KV Cache:

> **[System Prompt]** $\langle ct \rangle$
> **User:** Can you tell me where the garbage are when I drop them? **Observation:**
> [F][F]...[F][F][F]$\langle ct \rangle$ [HF]$\langle ct \rangle$ ...[F]$\langle ct \rangle$ [HF]$\langle ct \rangle$ **Assistant:** The garbage is dropped to the next to the notebook on the counter. $\langle ct \rangle$

Following this, solely the user's query content and the $\langle ct \rangle$ are maintained in the KV cache.

# D  Training Implementation Detail

In this section, we provide detailed implementation configurations of our training methodology.

**ESTP-Bench.** Our training methodology employs a three-stage strategy to progressively endow the VideoLLM-EyeWO model with advanced proactive capabilities. Tab. 6 summarizes key details of each stage's specific configuration and learning objectives, while Tab. 5 presents the corresponding ablation results.

**COIN-Benchmark.** Tab. 7 presents the training plan on the COIN [46] dataset. For a fair comparison with VideoLLM-Online [5], we utilized the same base model (LlaMa3 [16] and SigLIP [60]) and adopted identical training hyperparameters.

# E  ESTP-Bench Evaluation Details

In this section, we first present the parameter scales and hyperparameters for various models. Subsequently, we illustrate the corresponding prompts used in the evaluation.

| Data | Dataset | COIN |
|------|---------|------|
| **Model** | Base LLM | LLaMA3 |
| | Vision Encoder(s) | SigLIP |
| | Vision Token per Frame | Low Res: 1+3x3 |
| | | High Res: 1+7x7 |
| | Connector | MLP |
| | Trainable | Connector(full) |
| | | LLM(LoRA |
| | | r=128, scale=256) |
| **Initialization** | Connector | N/A |
| | LLM | LLaMA3 |
| **Training** | Batch Size per Device | 1 |
| | Gradient Accumulation | 8 |
| | Learning Rate | 1.5e-4 |
| | Warm-up Ratio | 0.05 |
| | LR Scheduler | Cosine |
| | Optimizer | Adamw |
| | Epochs | 5 |
| | Precision | bf16 fp16 |

Table 7: Training Plan in COIN [46]

| Model | Params | Frame | Query Frequency |
|-------|--------|-------|-----------------|
| *Offline MLLMs Response-in-Last* | | | |
| LLaVA-OneVision | 7B | 32 | Ask per Question |
| Qwen2-VL | 8B | 0.2-1 fps | Ask per Question |
| MiniCPM-V | 8B | 64 | Ask per Question |
| LLaVA-NeXT-Video | 7B | 32 | Ask per Question |
| InternVL-V2 | 8B | 32 | Ask per Question |
| *VLMs for Streaming Detection* | | | |
| CLIP | Base | 1 fps | 1 Hz |
| LaViLa | Base | 1 fps | 1 Hz |
| EgoVLP | Base | 1 fps | 1 Hz |
| *Offline MLLMs Polling Strategy* | | | |
| LLaVA-OneVision | 7B | 32 | 0.175 Hz |
| Qwen2-VL | 8B | 0.2-1 fps | 0.175 Hz |
| MiniCPM-V | 8B | 64 | 0.175 Hz |
| LLaVA-NeXT-Video | 7B | 32 | 0.175 Hz |
| InternVL-V2 | 8B | 32 | 0.175 Hz |
| *Online MLLMs* | | | |
| LIVE(threshold=0.8) | 8B | 2 fps | 2 Hz |
| LIVE(threshold=0.9) | 8B | 2 fps | 2 Hz |
| MMDuet | 8B | 2 fps | 2 Hz |
| VideoLLM-EyeWO(Ours) | 8B | 2 fps | 2 Hz |

Table 8: Parameter Scales and Hyperparameters for Various Models

### E.1 Various MLLMs Hyperparameter

**For Offline MLLMs**, considering Synchronized Efficiency, we employed models around the 7B/8B scale. For the Response-in-Last strategy, a single query was issued after processing the complete video segment. Regarding the Polling strategy, the querying frequency was set to $0.175$ queries per second. This frequency was chosen to match the model's response efficiency (Action Per Second), as illustrated in Fig. 7. Given that our videos are often significantly longer than the typical input frame capacity of these MLLMs, we followed the open-source code from [27], utilizing the corresponding sampling frequencies or input frame counts for different models.

**For VLMs**, we adopted model parameters and input sampling frequencies consistent with [13]. Given their inherent lack of text generation capabilities and consequently, their relatively low computational cost, the query frequency was set to match the input sampling frequency.

**For Online MLLMs**, we utilized model parameters, input frequencies, and query frequencies as reported in [5, 51].

## E.2 Evalutation Prompt

Below, we present the prompt used for the Offline MLLMs Response-in-Last strategy. Inspired by [37], we guided the model within the system prompt to output all answers corresponding to a given question, along with their respective frame indices. These frame indices are then converted into corresponding timestamps via a defined sampling relationship. Contextual information, encompassing both timestamps and content, is incorporated into the prompt following the approach of [27].

Below, we present the prompt used for the Offline MLLMs Polling Strategy, following the approach of [27]. The sole distinction lies in that proactive tasks in [27] solely require the model to judge if it can answer a question at the current moment, outputting only 'yes' or 'no'. In contrast, our approach additionally requires the model to generate a textual reply. Therefore, at timesteps where the model indicates readiness (by outputting 'yes'), we execute an additional query, prompting the model to generate the answer.

---

**Prompt Used for Response-in-Last**

```
You are an advanced video AI assistant. Given a video and a question, carefully
analyze each frame of the video, identify all relevant moments that help answer
the question, and provide the corresponding frame numbers along with the answer.
The format should be: '[frame idx] answer'.
For example, [6] The object is a cup.
[60] The object is a cup.
[100] The object is a yellow cup.
```

Here are the contextual information related to the video. Please answer the questions based on the contextual information:
At timestamp {}, the following question occurred: {}
At timestamp {}, the following answer occurred: {}

Here is the question. Answer it and don't confuse it with the previous conversation
Question: {}
The answer is:

---

**Prompt Used for Polling Strategy**

```
You are an advanced video question-answering AI assistant.  You have been
provided with video and a question related to the video.  Your task is to
carefully analyze the video and provide the answer to the question. You need
to carefully confirm whether the video content meet the conditions of the
question, and then output the correct content.
```

Here are the contextual information related to the video. Please answer the questions based on the contextual information:
At timestamp {}, the following question occurred: {}
At timestamp {}, the following answer occurred: {}

Here is the question. Answer it and don't confuse it with the previous conversation
Question: Is it the right time to answer the question "{}"? You need to answer yes or no. /
Please answer the question: "{}"
The answer is:

# F   Limitation and Future Work

To build a truly intelligent agent akin to Jarvis, it is essential to move beyond passive perception and incorporate active interaction with the environment. In this work, we focus on a constrained setting where the agent operates solely on pre-recorded videos. The objective is to train a model that can perceive the visual world and respond promptly within a fixed observation window. While this setting enables us to study perceptual grounding and response capabilities in a controlled manner, it inevitably limits the agent's ability to engage in real-time decision making, action planning, or closed-loop interaction.

This limitation highlights an important future direction: extending perceptual agents into embodied settings where they can interact with the environment through physical actions. For instance, integrating vision-language models into real-world robots-such as home assistants, warehouse manipulators, or autonomous drones—would require the ability to reason about action consequences, update beliefs based on new observations, and adapt behavior on the fly. Existing platforms like Habitat [32, 40, 45] for embodied AI or Meta's HomeAssist offer promising testbeds for such developments.

Therefore, future work should explore how to bridge the gap between static visual perception and dynamic agent-environment interaction. This includes developing methods for real-time perception-action loops, task-aware decision making, and cross-modal alignment under continual feedback. Strengthening interaction with the environment is a crucial step toward building general-purpose, autonomous agents that go beyond observation to perform useful actions in the real world.

# G Data Examples

## Task Type: Object Recognition

**Question:** What is the object on the desk near the tablet?

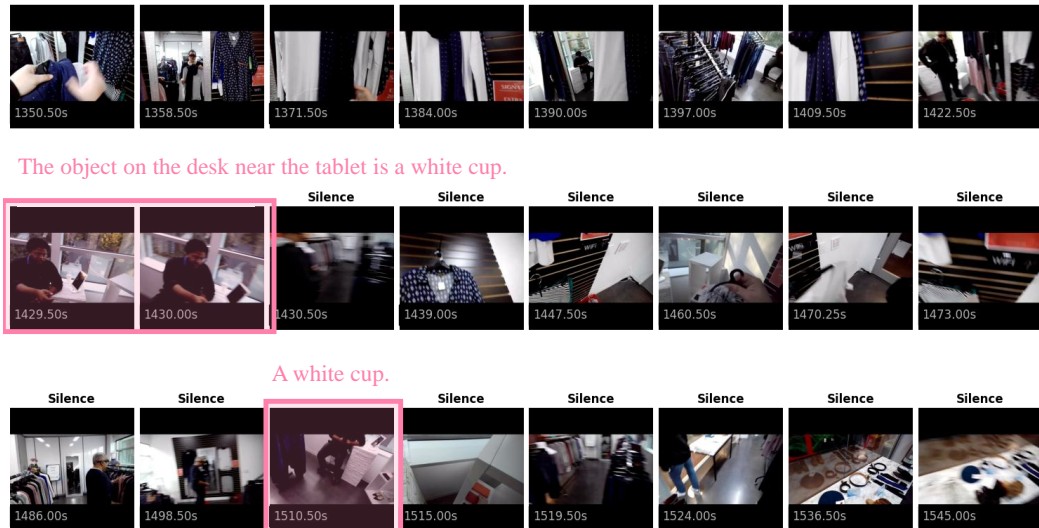

## Task Type: Attribution Perception

**Question:** What color is the frisbee held by the person in the frame?

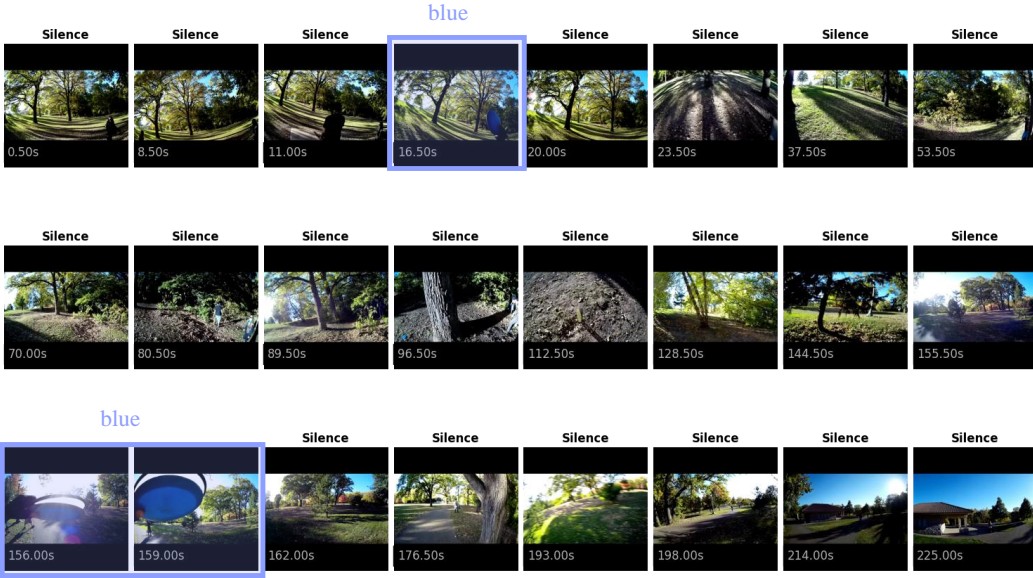

Figure 10: Data examples for Object Recognition, Attribution Perception tasks.

# Task Type: Text-Rich Understanding

**Question:** Can you remind me of the specific brand name written on the signboard above the store entrance?

# Task Type: Object Location

**Question:** Where's the blue bike's position in the room?

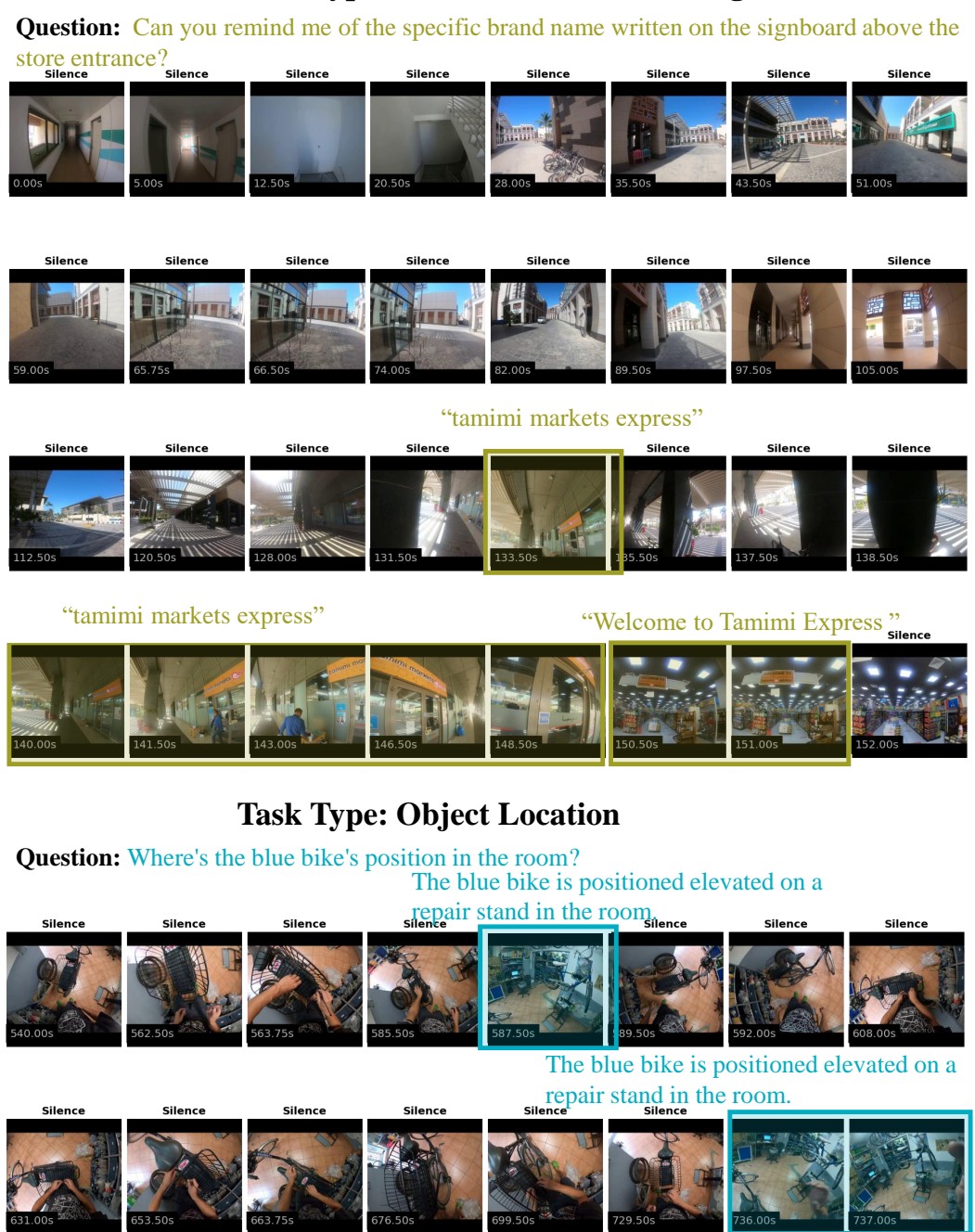

Figure 11: Data examples for Text-Rich Understanding, Object Location tasks.

## Task Type: Object State Change

**Question:** Can you remind me when the state of the fridge changes?

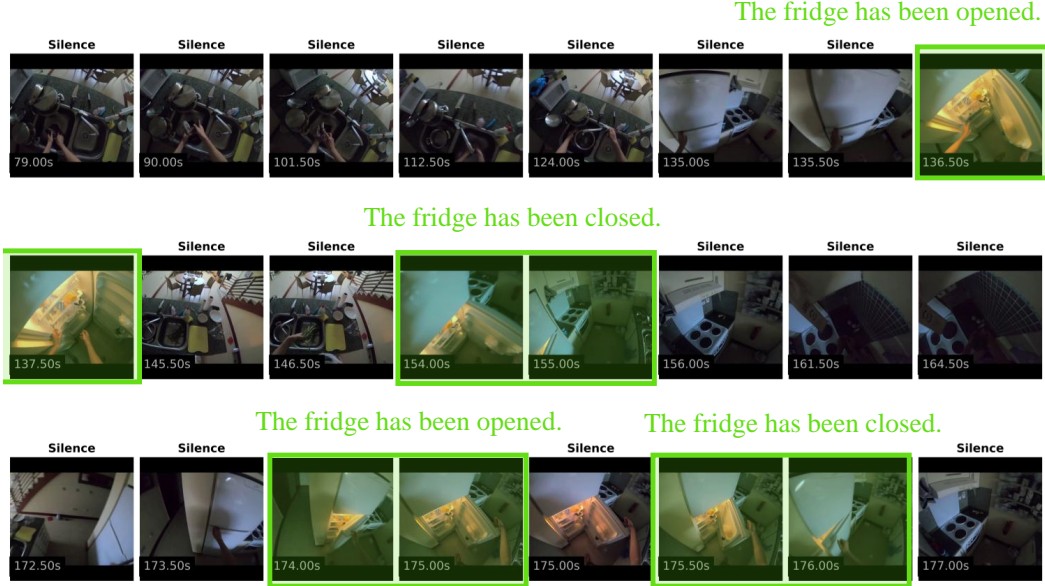

## Task Type: Ego-Object Localization

**Question:** How far the copper-colored pot is from me?

Figure 12: Data examples for Object State Change, Ego-Object Localization tasks.

# Task Type: Ego-Object State Change

**Question:** When does the tree change its position relative to me?

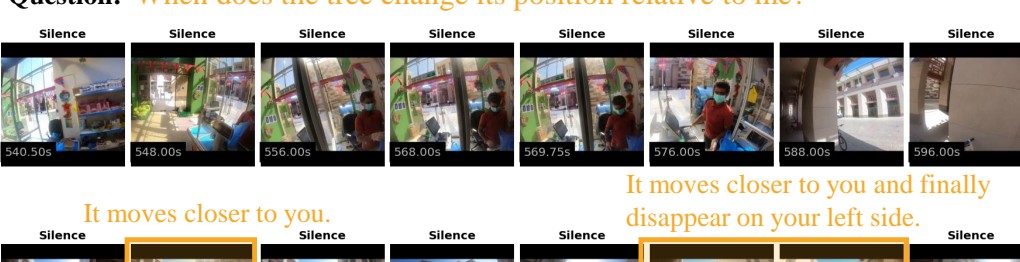

# Task Type: Action Recognition

**Question:** I want to know when I pick up items from the shelf in the market. Could you let me know?

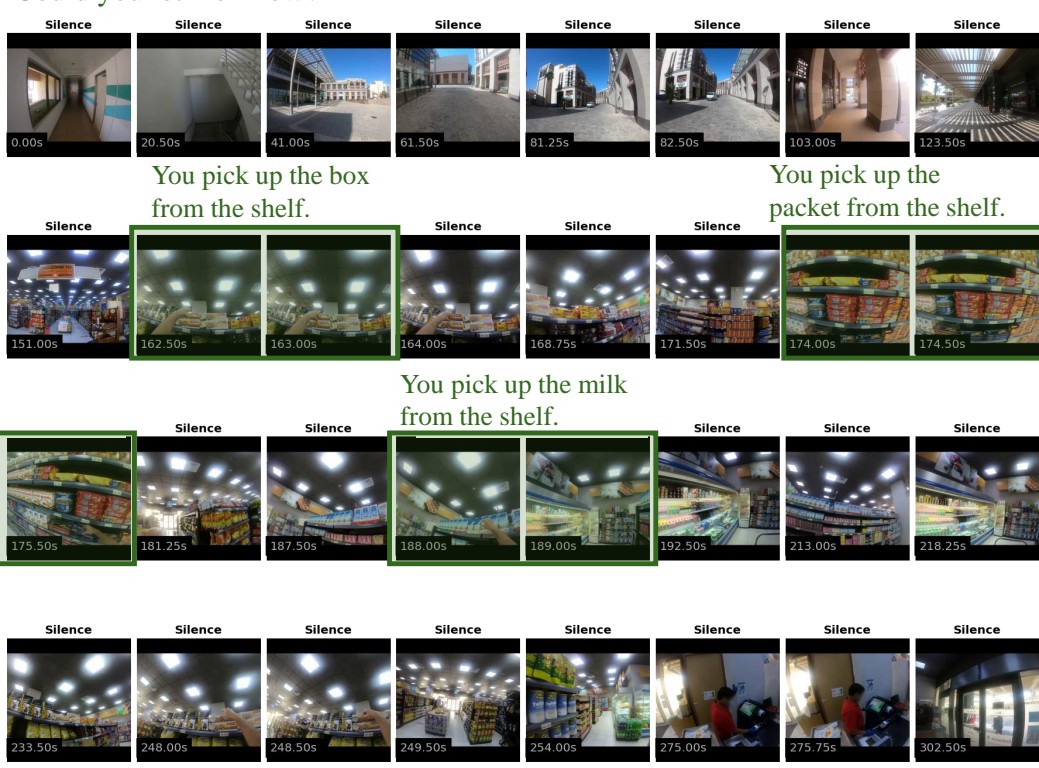

Figure 13: Data examples for Ego-Object State Change, Action Recognition tasks.

# Task Type: Object Function

**Question:** What could help if I want to load a lot of cargo to a far position?

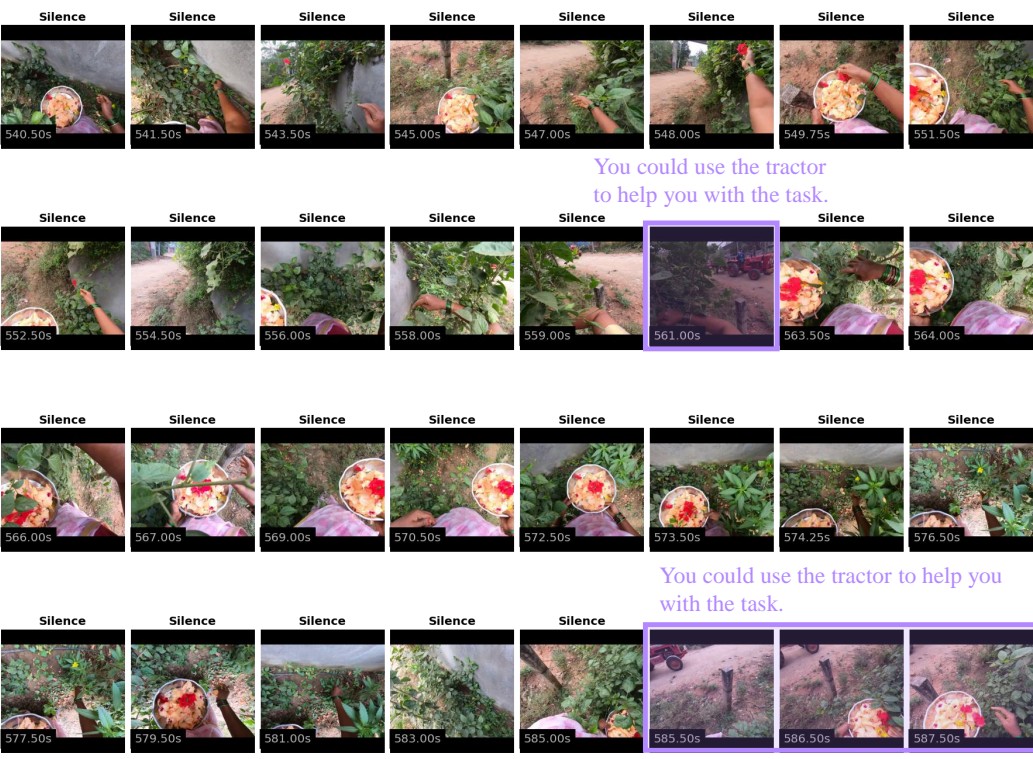

# Task Type: Information Function

**Question:** What time is it now?

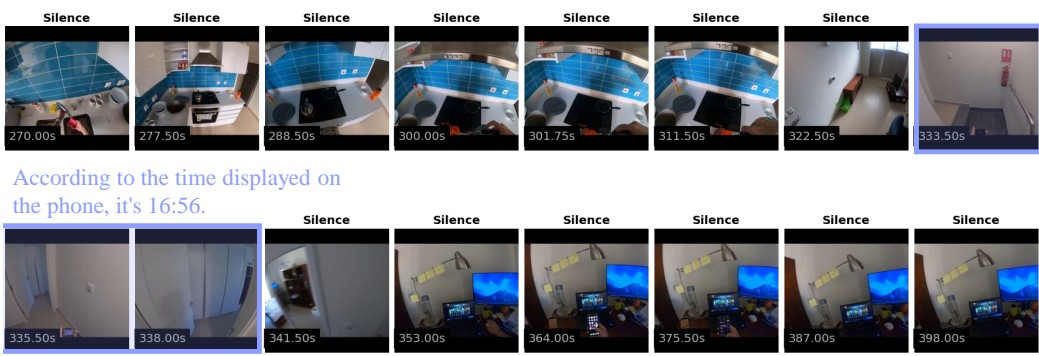

Figure 14: Data examples for Object Function, Information Function tasks.

# Task Type: Task Understanding

**Question:** Please explain how to achieve my goal of seasoning vegetables, starting from the very beginning and detailing each step.

To begin, pour lime juice over the vegetables. This will add a tangy flavor and help the seasoning adhere better.

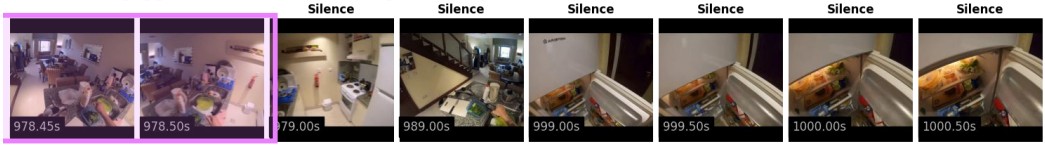

Next, add your chosen seasoning to the vegetables. This could include salt, pepper, herbs, or any other spices you prefer.

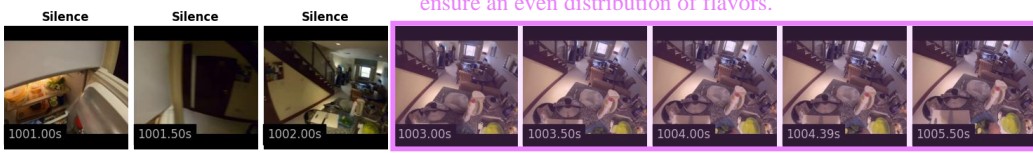

Finally, mix the seasoning thoroughly into the vegetables to ensure an even distribution of flavors.

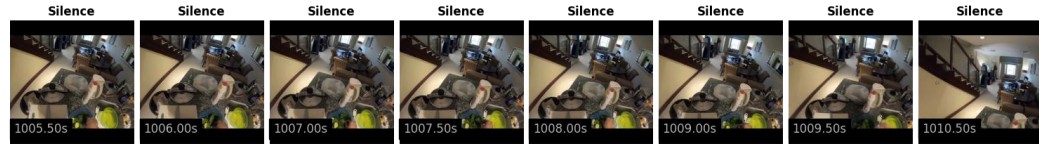

# Task Type: Next Action Reasoning

**Question:** What should I do next to continue dicing ingredients after completing the earlier tasks: washing utensils and slicing meat? Please provide clear instructions

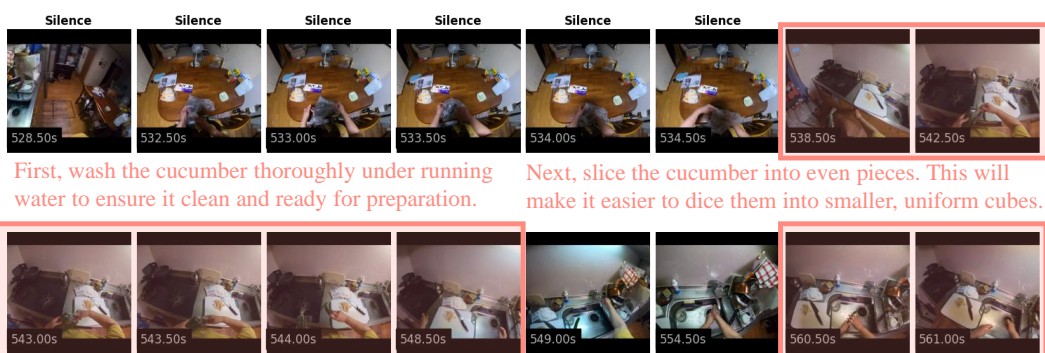

First, wash the cucumber thoroughly under running water to ensure it clean and ready for preparation.

Next, slice the cucumber into even pieces. This will make it easier to dice them into smaller, uniform cubes.

Figure 15: Data examples for Task Understanding, Next Action Reasoning tasks.

# Task Type: Object Relative Context

**Question:** [270s] Could you remind me when the dog is pooing?

**Question:** [332s] I want to clean it up. Give me some advice.

The dog is pooing.

You can use the green plastic bag in your right hand to collect them.

You can drop it into the dark bin on the left of you..

You can drop it into the dark bin on the left of you.

Figure 16: Data examples for Object Relative Context task.

# Task Type: Object Relative Context

**Question:** [98.1s] Can you analyze my current approach?

Currently, you're moving the mushrooms around in the pan, which is great for ensuring even cooking.

I noticed you're adjusting the heat this is a good practice to maintain control over the cooking process.

You've just added paprika spice, which will enhance the flavor profile.

Now that you've added chili flakes, the dish is gaining a nice kick of heat

**Question:** [150s] What are the resources I'll need for the next task?

For the next task, you'll need a spatula or spoon to continue moving the mushrooms, a heat source t maintain the cooking temperature, and possibly a timer to keep track of the cooking duration.

**Question:** [162.4s] What are the resources I'll need for the next task?

Your current goal is to cook the mushrooms thoroughly while ensuring the spices (oregano, paprika, and chili flakes) are evenly distributed.

You're now checking the recipe from the pack this is a good step to ensure you're on track.

Figure 17: Data examples for Task Relative Context task.

# H    Qualitative Results

**Question:  Where is the fire extinguisher located relative to the microwave?**

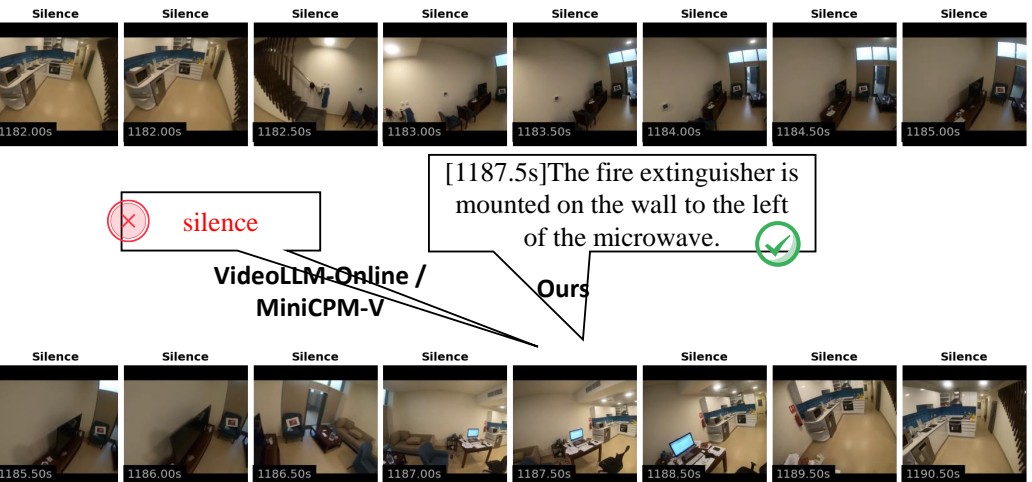

**Question:  Can you outline the subsequent steps for making a rice sushi roll after completing the previous steps: setting a nori sheet on the rolling mat, spreading rice on the nori sheet, and transferring rice from a rice cooker into the bowl with a spoon? Provide detailed guidance.**

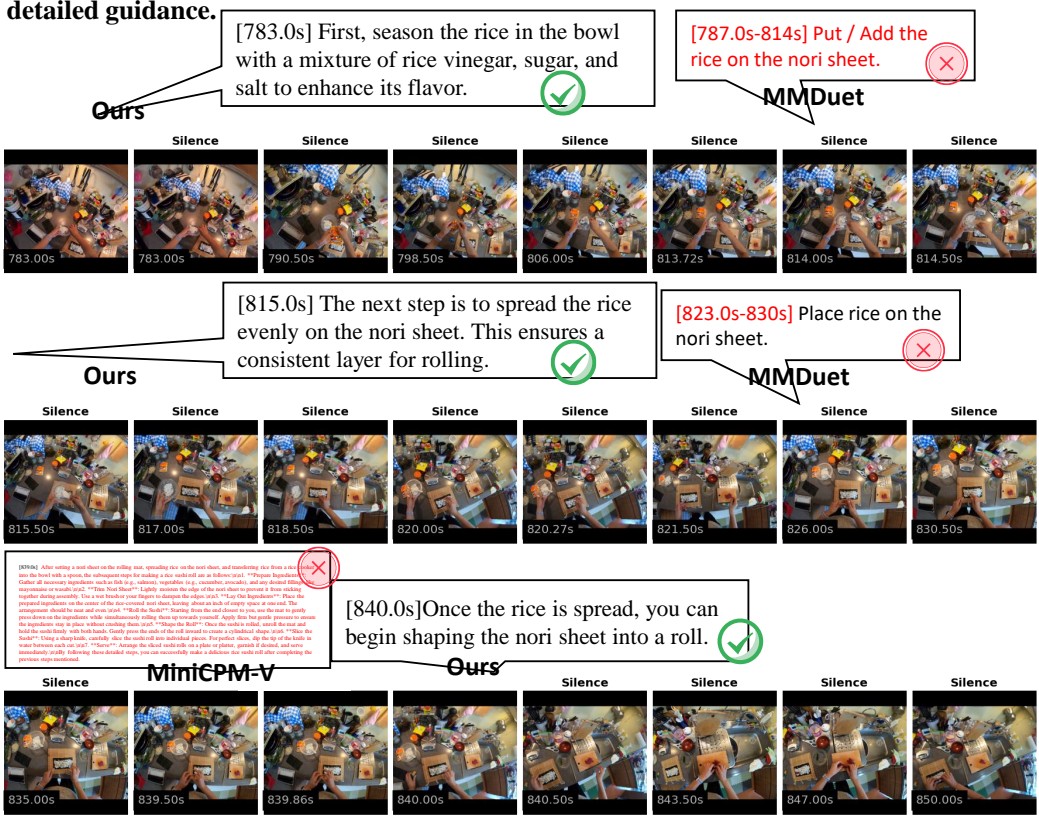

Figure 18: Qualitative Comparison with Baseline.

**Question: What can help in case of fire?**

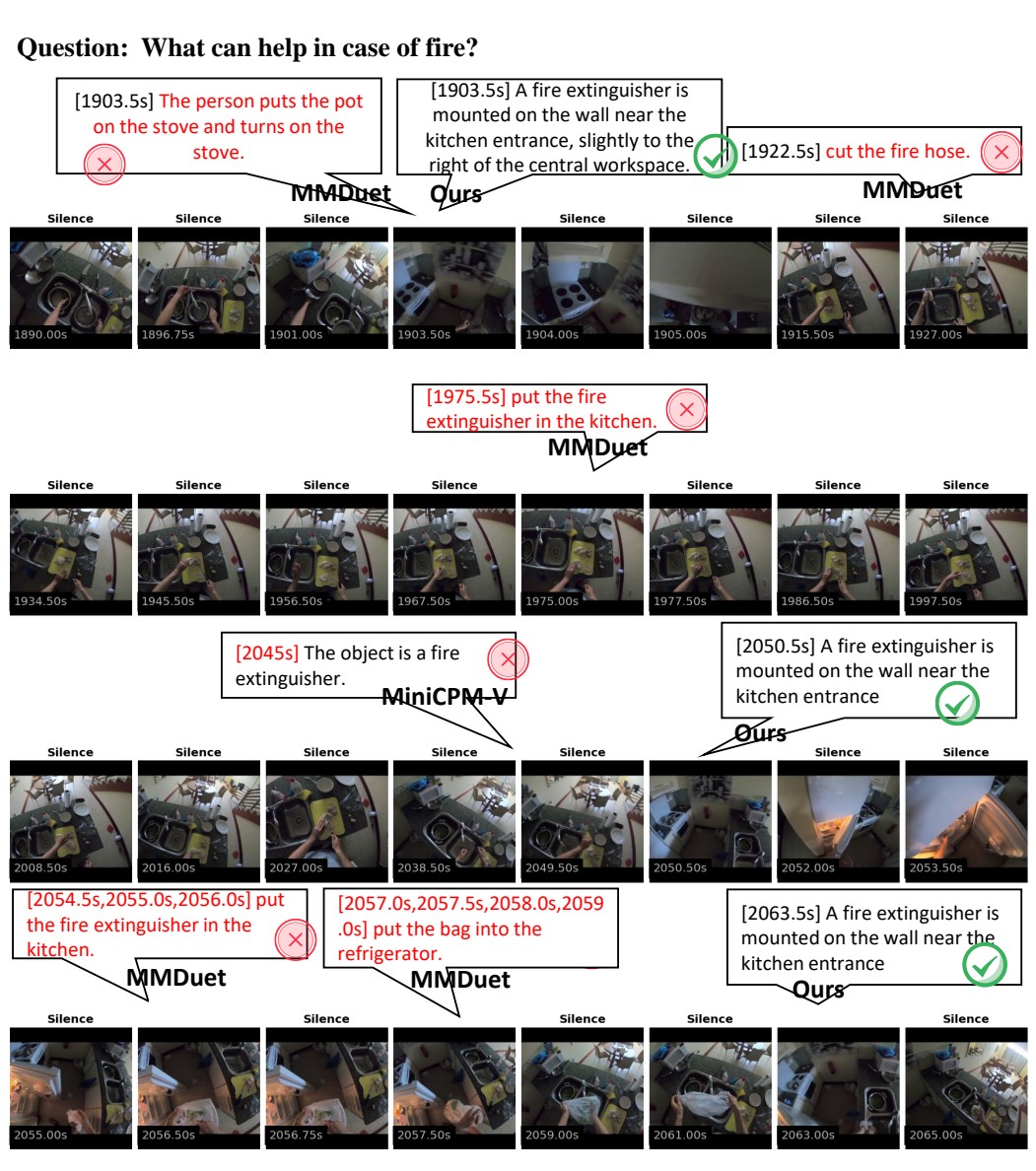

Figure 19: Qualitative Comparison with Baseline.