# OpenReview forum: "Eyes Wide Open: Ego Proactive Video-LLM for Streaming Video"
_NeurIPS.cc/2025/Conference — NeurIPS 2025 poster_

### Official Review · Reviewer_VD5g · 2025-06-22

**Clarity:** 2
**Significance:** 3
**Originality:** 3
**Rating:** 4
**Confidence:** 4

**Summary:**

The paper proposes a new task where, given ego-streaming video input, an assistant proactively answers questions while maintaining synchronized perception and reasoning. To enable this capability, the paper proposes an MLLM with calibrated benchmarks (ESTP-Bench) and metric (ESTP-F1). The model is trained in three stages and includes a dynamic compression technique to enable multi-turn QA while being efficient. The final model outperforms prior Online MLLMs as well as modified Offline MLLMs.

**Questions:**

My biggest concerns lie in the clarity. Please refer to the weakness section for more details.

One more question: any open-source plan for code and dataset?

**Ethical Concerns:**

["NO or VERY MINOR ethics concerns only"]

**Final Justification:**

My concerns are addressed. I hope the details/corrections will be incorporated in the future draft/revision.

**Limitations:**

no.

I would suggest the authors think of this carefully and add it in the revision. Possible limitations would include: how fast does the model run? Is it ready for real-time purpose? If so, on what kind of hardware device?

**Paper Formatting Concerns:**

no.

**Quality:**

3

**Strengths And Weaknesses:**

Strengths

1. Originality in terms of (1) task, (2) benchmark, and (3) loss design.

(1) The paper proposes a new task for MLLMs to proactively answer diverse given egocentric streaming videos.

(2) A new benchmark (ESTP-Bench) with a new metric that emphasizes (a) Proactive Coherence, (b) Just-in-Time Responsiveness, and (c) Synchronized Efficiency.

(3) A modified loss that addresses the training conflict due to the high similarity of adjacent frames in streaming inputs when using vanilla binary cross-entropy loss.

2. Significance of results.

(1) Clear improvement over prior Online MLLMs as well as modified Offline MLLMs.

(2) Generalizable on Offline Tasks (i.e., COIN benchmark).

Weaknesses

1. Notations are not always clear; sometimes inconsistent.

(1) L168, $\mathcal{T}\_\text{timestamp} $ = $ \mathcal{T}\_\text{GT} $? $ \mathcal{T}\_\text{GT} $ has never been used ever since.

(2)  Given the description, Eq (4) is simply a modulated version of Eq (3) with weighted supervision. $ \mathcal{T}\_\text{interval} $ should be the same as $\mathcal{T}\_\text{timestamp} $?

(3) Eq (5): $\mathcal{T}\_\text{uncertain} $ needs to be defined in the main text for completeness.

(4) Eq (6): It is $ O_t^h$  rather than $H_t$ that has to be highlighted.

(5) Eq (7): $\mathcal{L}\_\text{scrutinize}$ is never defined.

2. The two-level Compression makes sense. However, the implementation is not clear (L219 - L230). According to the paper, you are using a learnable special token \<ct\> to compress the input low-res multi-frame (or high-res single-frame). My question is, how do you supervised train this? How does the training enforce the compression? Some technical details and intuitions need to be added here.

3. "Triangle of Impossibility" in Figure 2 caption. This term refers to the trilemma where you cannot simultaneously achieve three goals. This contradicts the claim of the paper that the proposed method does well on all axes. I also believe that they are not essentially exclusive, and it's just a matter of dataset and model, which the paper's contribution lies in.

4. To get the number on COIN, what training data did you use? According to L149, the training dataset is Ego4D. If so, it is somewhat surprising to see that the model trained on egocentric videos does well on exocentric videos too.

5. Supplementary L130, since $\tau\_\text{ant} = 1$, $\tau\_\text{lat}=2$, $\text{scale}_k$ should always be greater than 3 even without $\max(1, \cdot)$?

---

> ### Author Rebuttal · Authors · 2025-07-28
>
> Thanks for your exceptionally thorough review and your detailed, positive summary of our work's originality and significance. We appreciate the precise suggestions for improvement, which we address in detail below.
>
> > **W1: Inconsistent notations**
> >
> 1. **W1.1:** Yes, in the LIVE training dataset, only answer timestamps are provided, not answer intervals. Therefore, $\mathcal{T}\_{timestamp}=\mathcal{T}\_{GT}$.
> 2. **W1.2:** No, we generate the ESTP-IT dataset, which includes answer interval annotations. To alleviate the conflicts that arise from timestamp-level supervision between adjacent frames, we incorporate interval-level temporal supervision signals. Concurrently, we employ a modified loss within these intervals to enable the model to discern subtle variations among frames within the given interval.
> 3. **W1.3:** Thank you for your suggestion. We provide a detailed definition of $\mathcal{T}\_{uncertain}$ in Appendix D (Stage-2 Input), and we'll revise the main text to include this information.
> 4. **W1.4:** Yes, thank you for pointing that out; we'll make the necessary revisions.
> 5. **W1.5:** Thank you for your correction. We'll amend this issue, as "scrutinize loss" in Eq. (7) refers to "ask_high loss" in Eq. (5).
>
> > **W2: Detail of Proactive Dynamic Compression**
> >
>
> For a multi-turn conversation, we forward **turn by turn**, inputting only the **system prompt, compression tokens from historical turns, and all tokens from the current turn**. This approach enforces compression during training. We provide an example in Appendix D Proactive Dynamic Compression Mechanism Example.
>
> > **W3: Caption for Figure 2**
> >
>
> Thank you for your suggestion. We will revise the caption for Figure 2, replacing "**Triangle of Impossibility**" with a term like "Multi-objective Trade-off Triangle" to enhance clarity. We also appreciate your recognition of our work's contribution.
>
> > **W4: COIN training set**
> >
>
> In our COIN experiments, we aim to demonstrate the **generalizability of our model architecture** (VideoLLM-EyeWO). Therefore, we follow the training setup of the baseline (VideoLLM-online), **also utilizing the COIN training set**. Thank you for your correction; we will subsequently supplement the main text with more detailed settings for COIN training.
>
> > **W5: Correction on $scale\_k$ Calculation**
> >
>
> Yes, thank you for your correction. Values without the maximum also exceed 3.
>
> > **Q1: Open-source plan**
> >
>
> We **plan to open-source the entire project**, including ESTP-Bench and VideoLLM-EyeWO (including training data and model weights). This effort aims to help researchers reproduce and build upon our work.
>
> > **L1: How fast does the model run? Is it ready for real-time purpose? If so, on what kind of hardware device?**
> >
>
> Our model, as depicted in Fig. 7, operates at an average rate exceeding **2 APS (Answers Per Second)** with a **2 FPS (Frames Per Second)** input. This experiment runs on an **8-A40 GPU cluster** configured with **96 Intel(R) Xeon(R) Gold 5318Y CPU @ 2.10GHz cores**. On our 8-H20 cluster, the running speed increases by 1.7 to 2 times. We appreciate your suggestion; further accelerating model execution and identifying suitable hardware setups remain meaningful challenges.

---

> ### Comment · Reviewer_VD5g · 2025-08-06
>
> I appreciate the author for providing the rebuttal. My concerns are addressed. I hope the details/corrections will be incorporated in the future draft/revision.

---

> > ### Author Response · Authors · 2025-08-06
> >
> > Thank you for your positive feedback and for your engagement throughout the review process! We are glad to hear that your concerns have been addressed. We confirm that all the details and corrections discussed will be carefully incorporated into the final revised version of the paper. If you have any further questions or require additional clarification, please do not hesitate to let us know.

---

### Official Review · Reviewer_jDVL · 2025-07-01

**Clarity:** 3
**Significance:** 3
**Originality:** 3
**Rating:** 4
**Confidence:** 4

**Summary:**

This paper defines the task of proactive question answering for AI assistants, using ego-streaming video as the input. The key properties emphasized are proactive coherence, just-in-time responsiveness, and synchronized efficiency. The authors construct a benchmark for this task, complete with an evaluation metric. They also propose a technical pipeline that addresses the challenge and demonstrates state-of-the-art (SOTA) results on both the proposed benchmark and a traditional video QA benchmark.

**Questions:**

1. The process for generating QA pairs and the corresponding response time annotations is not clearly explained in the main paper. Could you provide more details on how these annotations are generated?
2. In the ESTP-F1 metric, how are the components $S_{time}$ and $S_{answer}$ aggregated to obtain $S(g_k)$? A clearer explanation of this fusion process would be helpful.
3. Is there a more generalizable model solution that could address the various challenges mentioned in the task, beyond the current specific approach?

**Ethical Concerns:**

["NO or VERY MINOR ethics concerns only"]

**Final Justification:**

After the discussion with the authors during the rebuttal, my major concerns have been solved. Regarding the relatively limited novelty in model and metric design, I keep my score.

**Limitations:**

Not discussed in the main paper. Maybe limitations about the scalability and efficiency of training on more large-scale data can be discussed. Additionally, the scalability of the data generation process itself could be a concern.

**Quality:**

3

**Strengths And Weaknesses:**

Strengths
1. The paper introduces a novel proactive egocentric streaming video QA task aimed at helping developing AI assistants. The task design addresses key real-world application characteristics such as Proactive Coherence, Just-in-Time Responsiveness, and Synchronized Efficiency, making it highly valuable for research.
2. The authors propose a comprehensive benchmark and evaluation metric for the task, alongside a tailored solution to address the specific challenges of proactive video question answering. The results achieved are promising, and the work is thorough.

Weaknesses
1. The description of the proposed three-stage generation pipeline is insufficiently detailed. For instance, the use of RAG techniques for the one-to-many expansion is not clearly explained. Additionally, the integration of multi-turn QA and how to define the response time is unclear. Any manual annotations are needed? If needed, how to combine the automatically generated data and annotations is also not fully clarified. This lack of detail diminishes the perceived innovation in the dataset construction process and its potential for scalability.
2. The ESTP-F1 metric and the VideoLLM-EyeWO solution presented in the paper appear to have limited novelty. VideoLLM-EyeWO follows the framework of VideoLLM-online, with additional loss functions introduced at different stages to impose constraints on the model. However, these constraints are not exhaustive, which raises concerns about the scalability and generalizability of the approach.

---

> ### Author Rebuttal · Authors · 2025-07-31
>
> Thank you for your valuable feedback and for recognizing our contributions in establishing the ESTP task. To address your concerns, we provide detailed responses below that clarify the workings of our data generation pipeline and articulate the specific detail of my ESTP-F1 metric and VideoLLM-EyeWO model, distinguishing them from prior art.
>
> > **W1 and Q1: Data engine detail**
> >
>
> **W1.1 One-to-Many**
>
> In the **One-to-Many stage**, the core objective is to identify all segments within the captions that can answer the current question-answer (QA) pair. We explored two technical approaches:
>
> 1. **Inputting all captions into the Large Language Model (LLM)**: This method involves feeding the entirety of the captions into the LLM, tasking it with identifying all relevant segments and subsequently generating an answer. While this approach yields optimal results, it incurs substantial token consumption.
> 2. **Keyword-based Retrieval and Supplementary Generation**: During the one-to-one QA generation phase, we instruct the model to generate keywords pertaining to visual cues. These keywords are then used for retrieval, selecting relevant captions from the Multi-Perspective Caption. These selected captions are then input into the LLM, which is tasked with discerning their ability to answer the question and generating supplementary answers as needed.
>
> **W1.2 Integration of Multi-turn QA**
>
> Multi-turn QA is constructed by combining multiple single-turn QA pairs that pertain to the same video segment. Specifically, we leverage an LLM to identify logically related QA pairs and subsequently sequence them. Following this established order, the subsequent question is appended after a randomly selected number of preceding answers (one or more). This process is entirely rule-based and does not necessitate any manual annotation. Finally, an LLM is utilized to refine the generated text, ensuring it conforms to a fluent, streaming conversational style.
>
> **W1.3 Definition the Response Time**
>
> During training, response times are randomly sampled from the latter 50% of the answer interval. The generation of the answer interval itself is derived from the time intervals of relevant captions. The methodology for obtaining these caption intervals is detailed in **Appendix C.5: Multi-Perspective Caption Generation**. Briefly, for Event Captions, we follow the approach of EgoVLP for calculating interval start and end points. For Scene Captions, we leverage inter-frame similarity to determine potential interval boundaries.
>
> For evaluation, the optimal Response Time definition can be found in Appendix C.3.
>
> **W1.4 Manual Annotation**
>
> For ESTP-Bench, each data is manually annotated, with specific annotation details presented in Appendix C.2. For Data Engine and ESTP-IT, the data is entirely automatically generated without any manual annotation; more detailed generation processes can be found in Appendix C.4 and C.5.
>
> Thank you for your suggestion. We will subsequently add more details regarding data generation in the Appendix, such as how to prompt the LLM for RAG and the concatenation of multi-turn QA. Concurrently, we plan to open-source the data generation process.
>
> > **W2: ESTP-F1 metric and the VideoLLM-EyeWO**
> >
>
> **W2.1 ESTP-F1**
>
> Regarding the design of ESTP-F1, our aim was for it to be both simple yet comprehensive. We are the first to introduce F1 as a unified metric in streaming proactive tasks to simultaneously measure both response quality and timeliness. We note that reviewer Cx8y recognized the novelty of ESTP-F1, a point which was also appreciated by reviewer UR4Q.
>
> **W2.1 VideoLLM-EyeWO**
>
> VideoLLM-online, with its simple training and architecture, is our streaming proactive video LLM baseline. However, it falls short on **just-in-time responsiveness, accurate answers, and contextual understanding**—all essential for ESTP tasks. Our multi-stage training strategy, proactive dynamic compression, and data engine introduce these critical capabilities. These novel components are unique to our approach and are not found in prior proactive video LLM methods.
>
> We also agree that **scalability and generalizability** are critical.  To validate these aspects, we conduct additional experiments:
>
> **(1) Extended Benchmarking:** We test our model on other **online and offline video benchmarks**. This demonstrates its performance across a wider range of video understanding tasks.
>
> | Table 1 | **OvO-Bench Real-Time Visual Perception** | **OvO-Bench Backward Tracing** |
> | --- | --- | --- |
> | VideoLLM-online-8B | 20.79 | 17.73 |
> | Ours(VideoLLM-EyeWO-8B) | 32.76 | 28.00 |
>
> **Table 2: Performance comparison on the QaEgo4D test set.**
>
> | Table 2 | **QaEgo4D test-online** | **QaEgo4D test-offlline** |
> | --- | --- | --- |
> | VideoLLM-online-8B | 29.8 | 30.20 |
> | Ours(VideoLLM-EyeWO-8B) | 36.2 | 33.00 |
>
> **QaEgo4D test-online** evaluates the model's query performance in an online scenario, where inquiries are made *after* the answer interval appears. In contrast, **QaEgo4D test-offline** assesses performance when queries are made *after* the entire video has been loaded.
>
> **(2) Comparison with State-of-the-Art:**  We also incorporate the latest **LiveCC [1]** as a baseline. It's one of the most outstanding online video LLMs built on Qwen2-VL, and experimental results effectively showcase the efficacy of our method.
>
> | **Table 3** | **# Frames** | **OCR** | **ACR** | **ATR** | **STU** | **FPD** | **OJR** | **Avg.** | **EPM** | **ASI** | **HLD** | **Avg.** | **REC** | **SSR** | **CRR** | **Avg.** | **Overall Avg.** |
> | --- | --- | --- | --- | --- | --- | --- | --- | --- | --- | --- | --- | --- | --- | --- | --- | --- | --- |
> |  |  | **Real-Time** | **Visual** | **Perception** |  |  |  |  | **Backward** | **Tracing** |  |  | **Forward** | **Active** | **Responding** |  |  |
> | ***Open-source Multimodal Models-Offline*** |  |  |  |  |  |  |  |  |  |  |  |  |  |  |  |  |  |
> | Qwen2-VL-72B | 64 | 65.77 | **60.55** | 69.83 | 51.69 | 69.31 | 54.35 | 61.92 | 52.53 | **60.81** | **57.53** | **56.95** | **38.83** | 64.07 | 45.00 | 49.30 | **56.27** |
> | LLaVA-Video-7B | 64 | **69.13** | 58.72 | 68.83 | 49.44 | **74.26** | 59.78 | 63.52 | **56.23** | 57.43 | 7.53 | 40.4 | 34.10 | **69.95** | 60.42 | **54.82** | 52.91 |
> | LLaVA-OneVision-7B | 64 | 66.44 | 57.80 | **73.28** | **53.37** | 71.29 | **61.96** | **64.02** | 54.21 | 55.41 | 21.51 | 43.71 | 25.64 | 67.09 | 58.75 | 50.50 | 52.74 |
> | Qwen2-VL-7B | 64 | 60.40 | 50.46 | 56.03 | 47.19 | 66.34 | 55.43 | 55.98 | 47.81 | 35.48 | 56.08 | 46.46 | 31.66 | 65.82 | 48.75 | 48.74 | 50.39 |
> | InternVL-V2-8B | 64 | 67.11 | **60.55** | 63.79 | 46.07 | 68.32 | 56.52 | 60.39 | 48.15 | 57.43 | 24.73 | 43.44 | 26.5 | 59.14 | 54.14 | 46.60 | 50.15 |
> | LongVU-7B | 1fps | 53.69 | 53.21 | 62.93 | 47.75 | 68.32 | 59.78 | 57.61 | 40.74 | 59.46 | 4.84 | 35.01 | 12.18 | 69.48 | **60.83** | 47.50 | 46.71 |
> | ***Open-source Multimodal Models-Online*** |  |  |  |  |  |  |  |  |  |  |  |  |  |  |  |  |  |
> | Flash-VStream-7B | 1fps | 24.16 | 29.36 | 28.45 | 33.71 | 25.74 | 28.80 | 28.37 | 39.06 | 37.16 | 5.91 | 27.38 | 8.02 | 67.25 | 60.00 | 45.09 | 33.61 |
> | VideoLLM-online-8B | 2fps | 8.05 | 23.85 | 12.07 | 14.04 | 45.54 | 21.20 | 20.79 | 22.22 | 18.80 | 12.18 | 17.73 | - | - | - | - | - |
> | Dispider | 1fps | 57.72 | **49.54** | 62.07 | 44.94 | 61.39 | 51.63 | 54.55 | 48.48 | 55.41 | 4.3 | 36.06 | 18.05 | 37.36 | 48.75 | 34.72 | 41.78 |
> | LiveCC* | 2fps | 71.14 | 43.11 | 64.65 | 42.69 | 67.32 | 59.24 | 58.00 | 58.24 | 52.70 | **92.47** | 67.80 | 34.52 | 65.50 | 54.16 | 51.40 | 58.81 |
> | Ours (LiveCC-EyeWO) | 2fps | **78.52** | 44.95 | **71.55** | **45.50** | **69.30** | **63.04** | **62.14** | **59.60** | **60.13** | 89.78 | **69.83** | 32.80 | 64.22 | **64.58** | **53.87** | **62.00** |
>
> LiveCC*: We re-evaluate LiveCC's performance on our own devices.
>
> [1] LiveCC : Learning Video LLM with Streaming Speech Transcription at Scale
>
> > **Q2: ESTP-F1 Detail**
> >
>
> Specifically, we utilize an LLM to determine the score between the generated text and the ground truth, while also employing the temporal distance of timestamps to quantify response timeliness. Further details are comprehensively explained in Appendix C.3.
>
> > **Q3: VideoLLM-EyeWO shows strong potential to extend beyond its current tasks**
> >
> 1. VideoLLM-EyeWO is not a model tailored solely for the tasks proposed in our paper; instead, VideoLLM-EyeWO follows a general paradigm that supports adaptation to a wide range of tasks (see our response to W2.1).
> 2. The proactive task proposed in our paper is not only tailored for specific AI assistant scenarios, but also serves as a unified paradigm for conventional tasks. For example, it reduces to answering a single question given an image in typical image-based QA, and to performing short-range reasoning over a video clip in standard video QA. Therefore, a model like VideoLLM-EyeWO—designed to handle more generalized task formulations—holds strong potential for broader applications beyond those discussed in our paper.
>
> > **L1: VideoLLM-EyeWO shows strong Scaling Laws**
> >
> 1. **The model scales well with size, adapting effectively to various tasks.** For the newly proposed proactive task, we successfully trained a model of size 8B, demonstrating its effectiveness in handling the task and achieving promising results.
> 2. **The data scales well with size, with more data leading to significant performance improvements.** This is evident from our experiments, where increasing the dataset size consistently improved the model’s accuracy and generalization ability, particularly in complex tasks.
> 3. Since the data generation is fully automated, it allows for efficient scaling, ensuring a continuous supply of diverse and high-quality data without manual intervention.

---

> ### Comment · Reviewer_jDVL · 2025-08-06
>
> I appreciate the authors for providing the detailed responses. Most of my concerns have been addressed. The additional experiments on scalability and generalizability are convincing. I hope these results and details will be incorporated in the future draft or revision. I would like to maintain my concern regarding the novelty of the proposed models and metrics, but overall, it is a good effort to construct such an interesting benchmark. Therefore, I prefer to keep my original score.

---

> > ### Author Response · Authors · 2025-08-06
> >
> > Thank you very much for your thoughtful reply and for taking the time to review our rebuttal! We are glad to hear that our responses have addressed most of your concerns. We also sincerely appreciate your recognition of our efforts in constructing an interesting benchmark and your positive feedback on the convincing scalability and generalizability experiments.
> >
> > In the final version, we will ensure that all the additional results and details are carefully incorporated. We will also include a more thorough comparison and discussion with prior work to better highlight the unique contributions of our models and metrics.
> >
> > Thank you again for your constructive feedback, which is invaluable to improving our work! Please feel free to reach out if you have any further questions or suggestions.

---

> ### Comment · Area_Chair_svFD · 2025-08-08
>
> Dear Reviewer jDVL,
>
> This is the last call for author-reviewer discussion, which will ends on Aug 8. Could you please read authors' rebuttal and confirm whether your concerns have been addressed asap? Thank you.

---

### Official Review · Reviewer_Lpzn · 2025-07-01

**Clarity:** 3
**Significance:** 2
**Originality:** 2
**Rating:** 4
**Confidence:** 3

**Summary:**

This paper presents a compelling advancement in proactive video-language models, with a well-designed benchmark and innovative techniques. However, broader applicability, reproducibility, and statistical rigor could be strengthened. The work sets a strong foundation for future research in real-time, interactive AI systems.

**Questions:**

1. In Figure 7, why the model trained only on multi-turn QA (Stage 3) achieves higher APS compared to Stage 2?

2. In L212-218, the description of the compression is confusing. I could not understand (1) what are the two levels; (2) how many tokens are compressed; (3) what low/high compression rates did you choose.

**Ethical Concerns:**

["NO or VERY MINOR ethics concerns only"]

**Final Justification:**

Most of my concerns have been addressed by additional experiments. I consider slightly raise my score.

**Limitations:**

yes

**Quality:**

3

**Strengths And Weaknesses:**

Strengths:
1. The paper introduces a well-defined, challenging task (ESTP) that bridges the gap between passive observation and proactive AI assistance in streaming video scenarios.
2. The proposed benchmark rigorously evaluates the three key properties (proactivity, timeliness, synchronization) with diverse question types and contextual dependencies.


Weaknesses:
1. The "scrutinize loss" in Eq. (7) is not clearly referenced.

2. The authors should also report the performance of their proposed model on other online/streaming benchmarks, such as StreamingBench [1] and OVO-Bench [2], to prove its generalization ability.

3. In L79-L81, the authors claimed data engine as a key component in solving the ESTP task, however, I could not find more details about how this engine works except from L82-87.

4. The formatting of the references is completely incorrect; the authors should carefully review them before submission.

[1] Lin et al. StreamingBench: Assessing the gap for MLLMs to achieve streaming video understanding.

[2] Li et al. OVO-bench: How far is your video-LLMs from real-world online video understanding?

---

> ### Author Rebuttal · Authors · 2025-07-31
>
> Thank you for your valuable feedback and for acknowledging our contributions in defining the ESTP task. Following your suggestions, we have provided detailed clarifications on the "scrutinize loss" and data engine below, along with new experimental tables for OVO-Bench.
>
> > **W1: Typo of "scrutinize loss".**
>
> We apologize for the oversight; the term "**scrutinize loss**" in Equation (7) indeed refers to the "**ask_high loss**" introduced in Equation (5). We will rectify this inconsistency in subsequent revisions.
>
> > **W2: Performance on other online benchmarks.**
>
> Thank you for your valuable suggestion. We have conducted experiments on OvO-Bench to demonstrate our generalization capability. Concurrently, we selected VideoLLM-online as our baseline, given that it shares the same base model (LLaMA3 and SigLIP) and data source (Ego4D) as our own model.
>
> | Table 1            | Frames | OCR       | ACR       | ATR       | STU       | FPD       | OJR       | Avg.      | EPM       | ASI       | HLD       | Avg.      |
> | ------------------ | ------ | --------- | --------- | --------- | --------- | --------- | --------- | --------- | --------- | --------- | --------- | --------- |
> | VideoLLM-online-8B | 2fps   | 8.05      | 23.85     | 12.07     | 14.04     | **45.54** | 21.20     | 20.79     | 22.22     | 18.80     | **12.18** | 17.73     |
> | Ours (VideoLLM-EyeWO-8B)  | 2fps   | **24.16** | **27.52** | **31.89** | **32.58** | 44.55     | **35.87** | **32.76** | **39.06** | **38.51** | 6.45      | **28.00** |
>
> Compared to the baseline, our model demonstrates significant improvements. We also incorporate the latest **LiveCC [1]** as a baseline. It's one of the most outstanding online video LLMs built on Qwen2-VL, and experimental results effectively showcase the efficacy of our method.
>
> | Table 2                                     | # Frames | OCR           | ACR        | ATR            | STU       | FPD       | OJR       | Avg.      | EPM          | ASI         | HLD       | Avg.      | REC         | SSR        | CRR            | Avg.      | Overall Avg. |
> | ------------------------------------------- | -------- | ------------- | ---------- | -------------- | --------- | --------- | --------- | --------- | ------------ | ----------- | --------- | --------- | ----------- | ---------- | -------------- | --------- | ------------ |
> |                                             |          | **Real-Time** | **Visual** | **Perception** |           |           |           |           | **Backward** | **Tracing** |           |           | **Forward** | **Active** | **Responding** |           |              |
> | ***Open-source Multimodal Models-Offline*** |          |               |            |                |           |           |           |           |              |             |           |           |             |            |                |           |              |
> | Qwen2-VL-72B                                | 64       | 65.77         | **60.55**  | 69.83          | 51.69     | 69.31     | 54.35     | 61.92     | 52.53        | **60.81**   | **57.53** | **56.95** | **38.83**   | 64.07      | 45.00          | 49.30     | **56.27**    |
> | LLaVA-Video-7B                              | 64       | **69.13**     | 58.72      | 68.83          | 49.44     | **74.26** | 59.78     | 63.52     | **56.23**    | 57.43       | 7.53      | 40.4      | 34.10       | **69.95**  | 60.42          | **54.82** | 52.91        |
> | LLaVA-OneVision-7B                          | 64       | 66.44         | 57.80      | **73.28**      | **53.37** | 71.29     | **61.96** | **64.02** | 54.21        | 55.41       | 21.51     | 43.71     | 25.64       | 67.09      | 58.75          | 50.50     | 52.74        |
> | Qwen2-VL-7B                                 | 64       | 60.40         | 50.46      | 56.03          | 47.19     | 66.34     | 55.43     | 55.98     | 47.81        | 35.48       | 56.08     | 46.46     | 31.66       | 65.82      | 48.75          | 48.74     | 50.39        |
> | InternVL-V2-8B                              | 64       | 67.11         | **60.55**  | 63.79          | 46.07     | 68.32     | 56.52     | 60.39     | 48.15        | 57.43       | 24.73     | 43.44     | 26.5        | 59.14      | 54.14          | 46.60     | 50.15        |
> | LongVU-7B                                   | 1fps     | 53.69         | 53.21      | 62.93          | 47.75     | 68.32     | 59.78     | 57.61     | 40.74        | 59.46       | 4.84      | 35.01     | 12.18       | 69.48      | **60.83**      | 47.50     | 46.71        |
> | ***Open-source Multimodal Models-Online***  |          |               |            |                |           |           |           |           |              |             |           |           |             |            |                |           |              |
> | Flash-VStream-7B                            | 1fps     | 24.16         | 29.36      | 28.45          | 33.71     | 25.74     | 28.80     | 28.37     | 39.06        | 37.16       | 5.91      | 27.38     | 8.02        | 67.25      | 60.00          | 45.09     | 33.61        |
> | VideoLLM-online-8B                          | 2fps     | 8.05          | 23.85      | 12.07          | 14.04     | 45.54     | 21.20     | 20.79     | 22.22        | 18.80       | 12.18     | 17.73     | -           | -          | -              | -         | -            |
> | Dispider                                    | 1fps     | 57.72         | **49.54**  | 62.07          | 44.94     | 61.39     | 51.63     | 54.55     | 48.48        | 55.41       | 4.3       | 36.06     | 18.05       | 37.36      | 48.75          | 34.72     | 41.78        |
> | LiveCC*                                     | 2fps     | 71.14         | 43.11      | 64.65          | 42.69     | 67.32     | 59.24     | 58.00     | 58.24        | 52.70       | **92.47** | 67.80     | 34.52       | 65.50      | 54.16          | 51.40     | 58.81        |
> | Ours (LiveCC-EyeWO)                                | 2fps     | **78.52**     | 44.95      | **71.55**      | **45.50** | **69.30** | **63.04** | **62.14** | **59.60**    | **60.13**   | 89.78     | **69.83** | 32.80       | 64.22      | **64.58**      | **53.87** | **62.00**    |
>
> LiveCC*: We re-evaluate LiveCC's performance on our own devices.
>
> [1] LiveCC : Learning Video LLM with Streaming Speech Transcription at Scale
>
> > **W3: Data engine detail.**
>
> Due to main text page limitations, relevant details are provided in Appendix C.4 and C.5, with a framework diagram presented in Appendix Fig. 2.
>
> > **W4: Reference format error.**
>
> Thank you for pointing out this oversight. We sincerely apologize for not thoroughly checking this issue, which stems from our reference export process. We will correct all citation formats throughout the paper.
>
> > **Q1: Stage 3 and stage 2 APS comparison.**
>
> The KV Cache Size decreased after training with multi-turn QA (Stage 3) (from 3731.8 to 3242.8 as shown in Appendix Tab. 1). This reduction indicates that the model decreased the frequency of triggering responses. This, in turn, suggests that after multi-turn, more refined data training, the phenomenon of over-responding was reduced, leading to an improvement in APS.
>
> > **Q2: Detail of proactive dynamic compression.**
>
> **Q2.1 What are the two levels?**
>
> Given the inherent characteristics of streaming proactive tasks—specifically, that information at locations where the model initiates a request is more critical than information at silent locations—we assign a lower compression ratio to "request-triggering" locations (e.g., high-resolution frames, answers).
>
> **Q2.2 How many tokens are compressed?**
>
> As shown in Appendix Table 1, an average of 90% of tokens are compressed.
>
> **Q2.3 What low/high compression rates did you choose?**
>
> We do not choose specific, fixed numerical values for the low and high compression rates. Instead, the compression ratio is dynamic and relative, emerging naturally from our compression mechanism, which is designed for the unpredictability of streaming scenarios.
>
> Here is the principle:
>
> - Our method applies a **fixed number** of compression tokens to all regions, regardless of whether they are "request-triggering" or "silent."
> - **A relatively low compression rate** is inherently applied to "request-triggering" regions (such as high-resolution frames or answers). Because these critical regions contain significantly fewer original tokens, adding the fixed amount of compression tokens has a smaller relative impact, thus preserving their information with higher fidelity.
> - **A relatively high compression rate** is applied to "silent" regions. These regions are often long, containing many consecutive low-resolution frames. The same fixed number of compression tokens represents a more aggressive compression for these larger segments.
>
> This approach is crucial for streaming environments where the length of visual or textual inputs cannot be determined beforehand, as it allows the compression to adapt dynamically without relying on static, predefined ratios.

---

> > ### Author Response · Authors · 2025-08-07
> >
> > Dear Reviewer Lpzn,
> >
> > With the discussion period concluding in two days, we would be grateful for any further comments or suggestions to see if our responses have resolved your concerns.
> >
> > Thank you very much for your time and consideration.
> >
> > Sincerely,
> >
> > The Authors

---

### Official Review · Reviewer_UR4Q · 2025-07-01

**Clarity:** 3
**Significance:** 2
**Originality:** 2
**Rating:** 4
**Confidence:** 5

**Summary:**

The paper introduces ESTP-Bench, a benchmark for ego-centric streaming QA that emphasises (i) proactive coherence, (ii) just-in-time responsiveness and (iii) synchronised efficiency. It further proposes VideoLLM-EyeWO, a three-stage LoRA-based training pipeline with a data-generation engine, an action space that can request high-resolution frames, and a proactive compression token. On ESTP-Bench and offline tasks (COIN), EyeWO reportedly outperforms prior VLMs and VideoLLM variants.

**Questions:**

Please see weakness

**Ethical Concerns:**

["NO or VERY MINOR ethics concerns only"]

**Final Justification:**

Thank author for the detailed response. The additional rebuttal especially experiments from the other authors appears sufficient to me; therefore, I will raise my final rating.

**Limitations:**

Please see weakness

**Quality:**

2

**Strengths And Weaknesses:**

Strengths: (i) first introduce ESTP-Bench (Ego Streaming Proactive Benchmark) alongside the ESTP-F1 metric—a novel framework designed for their rigorous assessment. (ii) propose a comprehensive technical pipeline to enable models to tackle this challenging task.

Weaknesses:
i. Many components appear incremental: (a) the streaming ability is directly from LIVE.  (b) uniform compression resembles earlier token-compression work (Video-XL, VoCo-LLaMA). The data engine largely follows existing LVLM-based QA generation pipelines. Overall methodological novelty is limited.
ii. EyeWO is trained on ESTP-IT derived from Ego4D, whereas key baselines (VideoLLM-online, MMDuet) are evaluated zero-shot using released weights, as the authors explicitly state.  A fair comparison would (a) fine-tune those baselines on ESTP-IT or (b) report EyeWO in zero-shot.
iii. Experiments are only conducted on Ego4D-derived ESTP-Bench and COIN; no results are reported on other relevant ego-centric, streaming benchmarks (e.g., LIVE) or other offline datasets, making generalisation claims unsubstantiated.

---

> ### Author Rebuttal · Authors · 2025-07-31
>
> Thank you for your feedback and for recognizing the contributions of our new benchmark. To address your concerns, we have provided detailed responses below that further clarify the methodological novelty and present additional experimental results to support our claims regarding strong performance and generalizability.
>
> > **W1: Novelty**
>
> Both our **complete framework**—from data engineering to training and architecture (i.e., compression)—and **its components are novel, rather than mere incremental extensions** of prior work. The components mentioned by the reviewer cannot be readily adapted to ESTP, and straightforward combinations fail to address its challenges.
>
> We outline the core innovations of our framework, demonstrating how each part **meets the unique demands of ESTP** and why the mentioned components **fundamentally fall short**.
>
> | **Core Contribution**                                        | **Prior Approaches**                                         | **Inherent Challenge of ESTP**                               | **Limitation of Prior Approaches**                           | **Our Principle**                                            | **Our Proposed Solution and Result**                         |
> | ------------------------------------------------------------ | ------------------------------------------------------------ | ------------------------------------------------------------ | ------------------------------------------------------------ | ------------------------------------------------------------ | ------------------------------------------------------------ |
> | **1. Multi-Stage Training (Stage 1): Interval Responsiveness** | **LIVE:** Employs frame-level supervision and timestamp-based annotations. | while adjacent frames are nearly identical, the ground-truth label mandates a **sharp, binary shift from silence response.** | The training conflict results in a trade-off between over-responsiveness and under-responsiveness, **requiring manual tuning of response thresholds for suboptimal performance**. | To train the model to perceive subtle inter-frame changes by gradually transitioning the supervisory signal within an event-relevant interval, enabling a response only when sufficient information is aggregated. | **(1) Region-Based Modulation Loss:** Smooths the supervisory signal over an interval. **(2) Interval-annotated Data.** Achieves optimal performance without manual threshold tuning. *(See Ablation Study, Lines 2 & 3)* |
> | **2. Multi-Stage Training (Stage 2):** Ask High frame for just-in-time responsiveness and accurate answering. | **(1) VideoLLM-online:** Uses low-resolution frames exclusively.  **(2) Offline MLLMs (e.g., MM-Ego, VideoEspresso):** Select keyframes for accurate answering, but not in real-time. | (1) Low-resolution frames lack the spatial detail needed for decisive response timing and accurate answer generation.  (2) Inserting a high-resolution frame only *after* a response decision has been made does not aid the decision-making process itself. | Models struggle to achieve timely responsiveness and accurate answer generation. | The model learns to proactively request high-resolution details at key moments of interest. This enhanced information serves a dual purpose: first, to make a definitive judgment on whether a response is necessary, and second, to furnish the rich context required for an accurate and comprehensive answer. | The introduction of an `**ask_high**` **token**, trained in Stage 1, which enables the model to actively request high-resolution frames to improve both response timing and answer accuracy. *(See Ablation Study, Lines 4 & 5)* |
> | **3. Proactive and Dynamic Compression**                     | **(1) LLM-based compression (e.g., Video-XL):** Compresses single-turn visual inputs.  **(2) Input-stream compression (e.g., Flash-Vstream):** Merges frames before they reach the LLM. | (1) Single-turn compression schemes are ill-suited for multi-turn nature of streaming scenarios, which involve continuously **interleaved streams of visual and textual data.** (2) Pre-compressing the input stream prevents the LLM from perceiving every frame, hindering its decision-making. | (1) Compressing with a fixed-size window can interrupt ongoing reasoning or answer generation.  (2) Frame loss impairs the model's ability to learn when to respond. | (1) Each response turn forms a natural contextual boundary for compression.  (2) Information at moments that trigger a model request (e.g., for a high-res frame, answer) is more critical than at moments of silence. | **(1) Post-Response Compression:** Compression is only initiated after a response-generation cycle is complete, preventing interruption.  **(2) Two Level Compression Ratio:** A lower compression ratio is applied to more critical data segments (e.g., high-resolution frames, generated answers). |
> | **4. Data Engine**                                           | Existing data engine. (e.g **LLaVA-Next-Video** and **VideoEspresso**) | (1) Requires identifying all relevant moments for a potential question.  (2) Questions should target the very first appearance of an object or the start of an event.  (3) Data must support multi-turn, proactive dialogue. | (1) Incomplete recall of relevant moments can introduce training conflicts.  (2) Generated questions often pertain to persistent objects rather than novel events, failing to train true proactivity.  (3) Lack of multi-turn conversational flow. | (1)  identify key temporal moments by leveraging the event narrations from the Ego4D in conjunction with detected shifts in visual feature similarity. This allows us to precisely localize the scene changes, event occurrences, and initial object appearances. (2) Use an LLM to iteratively expand and consolidate these moments into coherent QA pairs. | Multi-Stage Data Engine and ESTP-IT. *(See Ablation Study, Lines 1 & 2)* |
> | **5. Holistic Multi-Stage Training Strategy**                | **Single-Stage Training**                                    | Different tasks exhibit a wide spectrum of complexity.       | Co-training disparate tasks leads to suboptimal performance, as the learning process for simpler skills can be compromised by more complex ones. | The training follows an "easy-to-hard" structure: it begins by establishing foundational responsiveness, then proceeds to requesting high-resolution information for precise answers, and concludes with fine-tuning for multi-turn dialogue. | The ablation studies in Tab Ablation demonstrate the effectiveness of each stage. |
>
> > **W2: Effect of ESTP-IT**
>
> In Appendix B, we conducted ablation experiments on ESTP-IT. As shown in Appendix Table 1, lines 1 and 2, ESTP-IT effectively improves the performance of VideoLLM-online. However, constrained by the training methodology, it was unable to achieve optimal performance. This further validates the effectiveness of our multi-stage training strategy.
>
> In the table below, we present the requested experimental results.
>
> | Table 1   | **Performance (Single Question) ↑** | **KV Cache Size (Single Question) ↓** | **Performance (Contextual Question) ↑** | **KV Cache Size (Contextual Question) ↓** |
> | --------- | ----------------------------------- | ------------------------------------- | --------------------------------------- | ----------------------------------------- |
> | LIVE      | 14.9                                | 9636.0                                | 18.9                                    | 31199.5                                   |
> | + ESTP-IT | 22.0                                | 7859.1                                | 25.7                                    | 28236.4                                   |
> | Ours      | **33.2**                            | **942.0**                             | **43.6**                                | **3242.8**                                |
>
> > **W3: **Limited discussion of model generalization****
>
> Thanks for the suggestion! We've expanded our evaluation to include more benchmarks and additional baselines.
>
> 1. **More benchmarks:**
>
> We select VideoLLM-online as a baseline due to its shared base model (LLaMA3 and SigLIP) and data source (Ego4D) with our VideoLLM-EyeWO. Our model consistently outperforms the baseline in both online and offline tasks, demonstrating strong generalization across diverse scenarios.
>
> - **OVO-Bench** for online video understanding.
>
>   | Table 2                  | **OvOBench Real-Time Visual Perception** | **OvOBench Backward Tracing** |
>   | ------------------------ | ---------------------------------------- | ----------------------------- |
>   | VideoLLM-online-8B       | 20.79                                    | 17.73                         |
>   | Ours (VideoLLM-EyeWO-8B) | 32.76                                    | 28.00                         |
>
> - **QaEgo4D** for offline video understanding.
>
>   | Table 3                  | **QaEgo4D test-online** | **QaEgo4D test-offline** |
>   | ------------------------ | ----------------------- | ------------------------ |
>   | VideoLLM-online-8B       | 29.8                    | 30.20                    |
>   | Ours (VideoLLM-EyeWO-8B) | 36.2                    | 33.00                    |
>
>   **QaEgo4D test-online** evaluates the model's query performance in an online scenario, where inquiries are made *after* the answer interval appears. In contrast, **QaEgo4D test-offline** assesses performance when queries are made *after* the entire video has been loaded.
>
> 2. **More baselines:** We adopt the most recent **LiveCC** [1] as our baseline. **Table 3** in our response to Reviewer jDVL consistently demonstrates the improvements achieved by our method, showcasing our model's strong generalization capability.
>
> [1] LiveCC : Learning Video LLM with Streaming Speech Transcription at Scale

---

> > ### Comment · Reviewer_UR4Q · 2025-08-06
> >
> > Thank author for the detailed response. I remain unconvinced by the comparison made to support the claimed novelty. Hope more technical efforts will be incorporated in the future revision.

---

> > > ### Author Response · Authors · 2025-08-06
> > > **Inquiry for the Clarification on Novelty Concern for Reviewer UR4Q**
> > >
> > > Thank you for your response. Aside from the novelty aspect, we would appreciate it if you could clarify whether there are any other concerns regarding our work.
> > >
> > > With respect to novelty, **we respectfully disagree with your assessment and would greatly appreciate more specific feedback.** In our view, the rebuttal have clearly articulated the fundamental differences and substantial innovations over prior work, both conceptually and empirically.
> > >
> > > - We would also like to emphasize that we are the **first** to introduce the **ESTP task**, along with a complete solution that spans from the benchmark (**ESTP-Bench**) and evaluation metric (**ESTP-F1**) to **a full methodological pipeline — covering data engineering, training strategy, and model architecture.**
> > > - Importantly, **neither the system-level contribution of this complete pipeline nor its individual components** can be effectively addressed or substituted by the methods mentioned by the reviewer, either conceptually or practically.
> > > - **Our results show clear and substantial improvements** over the reviewer-mentioned methods on ESTP-Bench, and strong performance across both online and offline video benchmarks, validating the effectiveness and generality of our approach.
> > > - If there are **particular aspects** you believe are insufficiently supported, we would be grateful if you could point them out so we can address them more directly.

---

### Official Review · Reviewer_Cx8y · 2025-07-02

**Clarity:** 3
**Significance:** 2
**Originality:** 2
**Rating:** 4
**Confidence:** 4

**Summary:**

This work introduces a novel task for an AI assistant that can proactively answer a wide range of evolving questions from an egocentric, streaming video input. The authors identify three key properties for this task: Proactive Coherence, Just-in-Time Responsiveness, and Synchronized Efficiency. To address these properties, the paper makes several contributions. First, it introduces the Ego Streaming Proactive Benchmark (ESTP-Bench) and a new evaluation metric, ESTP-F1, designed to assess these capabilities. Second, it proposes a comprehensive technical pipeline to enable models to perform this task. This pipeline includes a data engine for generating diverse, multi-turn questions and answers, a multi-stage training strategy to progressively build the model's capabilities, and a proactive dynamic compression technique to manage the growing volume of video data efficiently. The proposed model, VideoLLM-EyeWO, is shown to effectively address the challenges of the task and achieves SOTA performance on the COIN benchmark.

**Questions:**

1. The proactive dynamic compression mechanism is a key component of your proposed pipeline. Could you provide more details on how the model learns to determine when to compress and which compression level to apply? Is this a learned policy, and if so, how is it trained?
2. The paper mentions that the model can request high-resolution frames when it is uncertain. Could you elaborate on how the model's uncertainty is quantified and used to trigger this request?

**Ethical Concerns:**

["NO or VERY MINOR ethics concerns only"]

**Final Justification:**

My concerns have been solved well including the limited discussion of model generalization, the ask_high’s ablation study, basic settings of the method, etc. However, the complexity of the proposed method is still a problem, which may affect the reviewer's evaluation of the overall availability of the work.

**Limitations:**

Please see weakness.

**Paper Formatting Concerns:**

No concern.

**Quality:**

3

**Strengths And Weaknesses:**

Strength:
1. Novel and Important Problem Formulation: The paper tackles a timely and significant challenge in AI: moving from passive observation to proactive assistance in real-world, dynamic environments. The formulation of the Ego-Streaming Proactive (ESTP) task, with its emphasis on proactive, just-in-time, and synchronized responses, is a valuable contribution to the field.
2. Comprehensive Benchmark and Evaluation: The development of ESTP-Bench and the ESTP-F1 metric provides a much-needed framework for evaluating models on this complex task. The benchmark's inclusion of diverse question types (explicit, implicit, and contextual) and its focus on response timing and precision are particularly noteworthy.
3. Strong Empirical Results: The paper presents compelling experimental results, demonstrating the effectiveness of the proposed model, VideoLLM-EyeWO. The model's significant performance improvements over existing baselines on the ESTP-Bench and its state-of-the-art results on the COIN benchmark validate the proposed approach.

Weakness:

1. Limited Discussion of Generalization: While the paper demonstrates the model's performance on the Ego4D dataset, it would be beneficial to see a more in-depth discussion of how the proposed approach might generalize to other datasets and real-world scenarios. More evaluation results on online and offline video understanding and reasoning benchmarks are expected.
2. The main novelty of the framework seems to add an additional special token requesting a high-resolution frame. The ablation studies lack an analysis of this aspect.
3. Ablation Studies: While the paper presents a comprehensive evaluation, more detailed ablation studies could provide further insights into the specific contributions of each component of the proposed pipeline. For example, it would be interesting to see the impact of each stage of the training strategy or the effect of different compression levels.
4. Complexity of the Pipeline: The proposed technical pipeline is quite complex, with multiple components (data engine, multi-stage training, dynamic compression). This complexity could make it challenging for other researchers to reproduce and build upon the work.
5. The basic setting of the model VideoLLM-EyeWO is unclear. What is the input format for the video frames? Are they processed frame-by-frame or several frames together? How is the information from the historical trajectory stored and managed within the KV cache? What are the specific resolutions for the high and low-resolution frames?

---

> ### Author Rebuttal · Authors · 2025-07-29
>
> Thank you for your valuable suggestions for strengthening the evaluation and clarifying the technical details. We have provided detailed answers to each of your points below, supported by corresponding experimental tables.
>
> > **W1: Limited discussion of model generalization.**
> >
>
> Thanks for the suggestion! We've expanded our evaluation to include more benchmarks and additional baselines.
>
> 1. **More benchmarks:**
>
> We select VideoLLM-online as a baseline due to its shared base model (LLaMA3 and SigLIP) and data source (Ego4D) with our VideoLLM-EyeWO. Our model consistently outperforms the baseline in both online and offline tasks, demonstrating strong generalization across diverse scenarios.
>
> - **OVO-Bench** for online video understanding.
>
>
>     | Table 1 | **OvOBench Real-Time Visual Perception** | **OvOBench Backward Tracing** |
>     | --- | --- | --- |
>     | VideoLLM-online-8B | 20.79 | 17.73 |
>     | Ours (VideoLLM-EyeWO-8B) | 32.76 | 28.00 |
> - **QaEgo4D** for offline video understanding.
>
>
>     | Table 2 | **QaEgo4D test-online** | **QaEgo4D test-offline** |
>     | --- | --- | --- |
>     | VideoLLM-online-8B | 29.8 | 30.20 |
>     | Ours (VideoLLM-EyeWO-8B) | 36.2 | 33.00 |
>
>     **QaEgo4D test-online** evaluates the model's query performance in an online scenario, where inquiries are made *after* the answer interval appears. In contrast, **QaEgo4D test-offline** assesses performance when queries are made *after* the entire video has been loaded.
>
> - **COIN dataset** for offline temporal prediction.
>
>
>     | Table 3 | **Step** | **Task** | **Next** | **Proc** | **Proc+** |
>     | --- | --- | --- | --- | --- | --- |
>     | VideoLLM-online-8B | 63.1 | 92.6 | 49.0 | 49.7 | 53.6 |
>     | VideoLLM-MOD | 63.4 | 92.7 | 49.8 | 49.8 | 53.3 |
>     | Ours | 65.9 | 92.7 | 50.9 | 50.8 | 54.7 |
>
> 2. **More baselines:** We adopt the most recent **LiveCC** [1] as our baseline. **Table 3** in our response to Reviewer jDVL consistently demonstrates the improvements achieved by our method, showcasing our model's strong generalization capability.
>
> [1] LiveCC : Learning Video LLM with Streaming Speech Transcription at Scale
>
> > **W2: The ask_high’s ablation study and other key novelty.**
> >
> - Indeed, **ask_high** is one of our core novelties, making it highly suitable for streaming and proactive scenarios. It processes key frames at high resolution for frame-by-frame input, providing crucial information for response decision-making and answer generation. Table 4 (lines 4 and 5) further confirms its critical role in these processes.
> - Beyond ask_high, our framework introduces several other significant core contributions:
>     1. **Interval Responsiveness:** A new training method and loss function to handle the "silence to response" shift without manual thresholds (in Sec 3.2 Stage-1). Without this, a trade-off between over- and under-responsiveness emerges, requiring manual tuning and leading to suboptimal performance. Unlike the abrupt supervisory signals in LIVE, our method smooths the signal over an interval, fostering a more stable and effective training process.
>     2. **Proactive Dynamic Compression :** A novel compression strategy tailored for streaming, multi-turn scenarios (in Sec 3.3). Traditional compression fails in these contexts because single-turn schemes don't suit interleaved visual/textual streams, and pre-compressing prevents the LLM from perceiving every frame, hindering decision-making.
>     3. **Data Engine:** A new multi-stage pipeline for generating high-quality, proactive training data. Unlike prior video QA, proactive task requires just-in-time and contextual information. Our framework automatically derives captions with time interval for a video , then generating QA pairs for critical segments, finally completing answers and merging questions via captioning.
>     4. **Holistic Multi-Stage Strategy :** The curriculum learning approach itself, which outperforms single-stage training.
>
> > **W3: Ablation studies.**
> >
> - Our ablation studies, detailed in the table below and Appendix B, confirm the significant impact of **ask_high** (Table lines 4-5). These studies also highlight the crucial roles of **ESTP-IT**, **multi-stage training**, and the **proactive dynamic compression mechanism**.
> - As suggested, the Table 4 also includes **KV Cache Size statistics for dynamic compression** (lines 3-4), demonstrating a significant reduction without performance compromise.
>
> **Table 4: Ablation Study on ESTP Bench.**
>
> | **Index** | **Method** | **Performance (Single Question) ↑** | **KV Cache Size (Single Question) ↓** | **Performance (Contextual Question) ↑** | **KV Cache Size (Contextual Question) ↓** |
> | --- | --- | --- | --- | --- | --- |
> | 1 | LIVE | 14.9 | 9636.0 | 18.9 | 31199.5 |
> | 2 | + ESTP-IT | 22.0 | 7859.1 | 25.7 | 28236.4 |
> | 3 | Stage-0 | 24.9 | 7988.2 | 23.0 | 17567.6 |
> |  | *with increased proactive dynamic  compression mechanism* |  |  |  |  |
> | 4 | + Stage-1 ask low frame | 27.3 | 623.1 | 25.1 | 1592.3 |
> | 5 | + Stage-1 ask high frame | **34.0** | 1182.8 | 38.7 | 3731.8 |
> | 6 | + Stage-2 | 33.2 | 942.0 | **43.6** | 3242.8 |
>
> > **W4: Complexity of the pipeline.**
> >
>
> Our pipeline comprehensively explores the ESTP task from three distinct yet interconnected perspectives: **data, training, and architectural design**. While each angle offers a unique focus, they share a common, intuitive design motivation: to efficiently and accurately answer questions at precise moments within a streaming scenario.
>
> We intend to open-source the entire project, including ESTP-Bench, VideoLLM-EyeWO (encompassing training data and model weights), to facilitate reproducibility and further research within the academic community.
>
> > **W5: Basic settings.**
> >
> 1. **W5.1 Input Format** : The model processes input **frame by frame at 2 FPS**. After each frame, the model outputs a decision: continue to the next frame or generate a response. (See Appendix D for a LIVE Input example.)
> 2. **W5.2 historical trajectory storage:** Specifically, after the model decides to respond and generate an answer, Proactive Dynamic Compression is executed, storing only compressed, compact tokens in the KV cache. We also keep the System Prompt at the very beginning of the KV cache. (Appendix D, "Proactive Dynamic Compression Mechanism Example," offers more detail.)
> 3. **W5.3 Specific Resolution**: Following SigLIP, our input frames use a **384x384 resolution**. Low-resolution frames are pooled into 10 tokens (1+3x3), consistent with VideoLLM-online, while high-resolution frames are represented by 50 tokens (1+7x7). (Appendix E provides more detail.)
>
> > **Q1: Training of proactive dynamic compression.**
> >
>
> **Q1.1: How the model learns to determine when to compress and which compression level to apply?**
>
> Given the nature of streaming proactive tasks—(1) where each response turn naturally segments context and (2) information from "request-triggering" positions is more crucial than from "silent" ones—we employ the following strategy:
>
> 1. **Compression is performed after a response is completed.** This approach prevents interruption of ongoing text generation or the response judgment during frame input.
> 2. **Equal amounts of compression tokens are appended to both "request-triggering" regions (e.g., high-resolution frames, answers) and "silent" regions.** This inherently results in a **dynamic compression ratio**: lower for request-triggering regions and higher for silent ones, as the original token count in request-triggering regions is significantly smaller.
>
> This mechanism, combined with the model's proactive behavior in streaming tasks, imbues the model with the ability to perform proactive, dynamic compression through training on the original task.
>
> **Q1.2:  How is it trained?**
>
> For a multi-turn conversation, we forward **turn by turn**, inputting only the **system prompt, compression tokens from historical turns, and all tokens from the current turn**. This approach enforces compression during training. We provide an example in Appendix D Proactive Dynamic Compression Mechanism Example.
>
> > **Q2: Quantify uncertain interval.**
> >
>
> We define the **model's erroneously triggered response regions** as uncertain intervals. Specifically, we first train for one epoch using the annotated answer intervals as the initial uncertain intervals. Subsequently, we perform an inference pass over the data, identifying positions where high-resolution requests occur outside the annotated intervals and designating these as uncertain intervals. This process, along with input examples, is detailed in Appendix D Stage-2 Input.

---

> > ### Comment · Reviewer_Cx8y · 2025-08-05
> > **Thank you for the response**
> >
> > Thank the authors fot the detailed response. My concerns have been sovled well and I will consider to improve my rating score. Furthermore, I hope authors to provide more implementation and experimental details in the next version.

---

> ### Author Response · Authors · 2025-08-05
>
> Thank you for your timely response! We are pleased to hear that our rebuttal has addressed your concerns well, and we appreciate your consideration of improving the rating. We will carefully revise the paper in accordance with your valuable suggestions, including (but not limited to) adding experiments presented in the rebuttal to further evaluate model generalization on additional online and offline benchmarks, incorporating more baselines, and conducting extended ablation studies. We will also improve the clarity of the implementation and experimental details.
>
> In response to your further suggestion for clarification on the implementation and experimental setups, we provide a concise summary below. More detailed information will be added to the main paper:
>
> - VideoLLM-Eye: Detailed training implementation is provided in Appendix E: Training Implementation Detail of our submission.
> - OvObench and QaEgo4D: Evaluations on these benchmarks were conducted under zero-shot testing.
> - LiveCC-EyeWO: We fine-tuned the LiveCC model using LoRA. The training data consists of the ESTP-IT 80K dataset and 30K QA pairs randomly sampled (due to time constraints) from the LLaVA-Video-137k dataset, which is part of LiveCC’s original training set.
>
> In addition, **we will open-source the entire project,** including **ESTP-Bench** and **VideoLLM-EyeWO,** covering **training data, implement details**, and **model weights.**
>
> Thank you again for your constructive feedback and consideration to raise our paper’s rating!

---

> ### Comment · Area_Chair_svFD · 2025-08-08
>
> Dear Reviewer Cx8y,
>
> This is the last call for author-reviewer discussion, which will ends on Aug 8. Could you please read authors' rebuttal and confirm whether your concerns have been addressed asap? Thank you.

---

### Comment · Area_Chair_svFD · 2025-08-04

Dear Reviewers,

This paper received a range of ratings prior to the rebuttal i.e., Borderline Reject / Borderline Accept / Reject.

Now that the author response is available, please take this opportunity to assess whether your concerns have been adequately addressed. If not, feel free to request clarifications or engage further with the authors.

---

### Note · Authors · 2025-08-12

Dear ACs and Reviewers,

Thank you for your valuable feedback throughout the review process. We appreciate the opportunity to engage in discussion and to now provide a summary.

**Our Contributions:**
1. **A Novel and Important Task:** We propose the innovative ESTP task, unanimously recognized by **all reviewers** as novel. Cx8y and jDVL highlight its importance for real-world applications.
2. **A Comprehensive Benchmark and Metric:** We are the **first** to introduce ESTP-Bench and ESTP-F1, an evaluation framework praised by Cx8y and jDVL as comprehensive and much-needed, and acknowledged for novelty by VD5g and UR4Q.
3. **An Effective Technical Pipeline with Strong Performance:** We propose a comprehensive technical method to address ESTP task and online video understanding challenges. UR4Q praised the pipeline's comprehensive. The strong results are validated by reviewers—Cx8y calls them compelling, jDVL finds them promising and thorough, and Lpzn highlights the innovative techniques.

**Addressing Key Points in Discussion Phase:**
- **Generalization and Scalability of Our Method Demonstrated:** We expand evaluations to both **online** (OvO-Bench) and **offline** (COIN, QaEgo4D) benchmarks, adding a **SoTA baseline** (LiveCC). UR4Q and Lpzn acknowledge this; Cx8y and jDVL confirm their concerns are resolved.
- **Ablation Studies and Clarifications:** We provide additional ablation studies and methodological details. Cx8y, jDVL and VD5g confirm that their concerns are well-solved. We will **open-source the entire project**.
- **Summary of Our Method’s Novelty for UR4Q:**
    1. Our method is a **system-level contribution**, comprising **a data engine, a multi-stage training paradigm, and a model architecture**—the **first complete pipeline** for ESTP.
    2. **Each pipeline component is also novel and clearly distinct from prior work**. As detailed in our response, existing methods are **fundamentally unsuited** for the unique demands of ESTP task. Our innovations—from the interval-based training strategy to the proactive dynamic compression—offer new perspectives and technical advances for online VideoLLMs and the specific ESTP task, both **conceptually and practically**.
    3. Our method represents **only part of our technical contribution.** The paper's novelty also lies in **being the first to define the ESTP task,** while also introducing a **multi-faceted benchmark** and **evaluation** metric to drive progress.

Sincerely,

The Authors

---

### Decision · Program_Chairs · 2025-09-17

**Decision:**

Accept (poster)

**Comment:**

This paper introduces a proactive ego-centric Video-LLM framework for streaming video understanding, enabling real-time processing through a novel mechanism that anticipates future frames while maintaining temporal coherence.

Reviewers agree that the work addresses an important and timely problem of efficient streaming video understanding, offering strong technical contributions, solid experiments, and meaningful improvements over existing baselines.

Strengths include the originality of the proactive streaming design, clear motivation, and comprehensive evaluation across diverse tasks. Weaknesses mainly concern limited analysis of scalability and ablation depth, as well as the need for clearer articulation of broader applicability beyond ego-centric scenarios, which should be included in a revision.

The rebuttal period saw constructive responses that addressed concerns to a reasonable extent.  Finally, all reviewers voted for accept, though they suggested clarifying broader impact and extending empirical validation.  Given the novelty, rigor, and potential impact on the growing field of streaming multimodal agents, I recommend acceptance (poster).